# Bactericidal membrane attack complex formation initiates at the new pole of *E. coli*

Marije F L van 't Wout [ID][1], Fabian Hauser [ID][2], Philippa I P Holzapfel [ID][1], Bart W Bardoel[1], Carla J C de Haas[1], Jaroslaw Jacak[2], Suzan H M Rooijakkers [ID][1] & Dani A C Heesterbeek [ID][1✉]

## Abstract

**Human immune protection against bacteria critically depends on activation of the complement system. The direct bacteriolytic activity of complement molecules against Gram-negative bacteria acts via the formation of Membrane Attack Complex (MAC) pores. Bactericidal MAC pores damage the bacterial outer membrane, leading to destabilization of the inner membrane. Although it is well-established that inner membrane damage is crucial for bacterial cell death, the critical event causing MAC-mediated inner membrane damage remains elusive. Here we question whether the bacterial cell envelope possesses vulnerable spots for MAC pores to insert. By following the localization of MAC pores on *E. coli* over time using fluorescence microscopy, we elucidate that MAC deposition initiates at the new bacterial pole, which induces inner membrane damage and halts bacterial division. MAC components C8 and C9 preferentially localize at new bacterial poles, while C3b localizes randomly on the bacterial surface. This suggests that preferential MAC localization is determined by one of the initial steps of MAC formation. These findings provide valuable information about the interplay between immune components and the Gram-negative cell envelope.**

**Keywords** Membrane Attack Complex; Complement System; Gram-negative Bacteria; Fluorescence Microscopy; Image Analysis
**Subject Categories** Cell Adhesion, Polarity & Cytoskeleton; Immunology; Microbiology, Virology & Host Pathogen Interaction

## Introduction

The immune system protects against bacterial infections by recognizing and attacking bacteria that invade the human body. One of the first lines of defense against pathogenic bacteria is the complement system, which consists of a large family of proteins in the blood and other body fluids (Dunkelberger and Song, 2010). The importance of this system in the defense against bacteria is illustrated by the recurrent infections that occur in people with complement deficiencies (Ram et al, 2010; Turley et al, 2015) or patients treated with complement inhibitors (Winthrop et al, 2018). Complement proteins can become activated upon recognition of bacteria, starting a cascade of cleavage reactions on the bacterial surface. Complement activation triggers multiple effector mechanisms that aid in bacterial clearance, one of them being the formation of Membrane Attack Complex (MAC) pores. MAC pores are formed when C5 convertase enzymes on the bacterial surface generate C5b, which can assemble with C6, C7, C8 and up to 18 copies of C9 to form a complete MAC pore (Parsons et al, 2019). By causing large-scale disruption of the bacterial cell envelope, MAC pores can directly kill Gram-negative bacteria (Benn et al, 2024; Doorduijn et al, 2019).

Gram-negative bacteria possess a complex cell envelope including both an inner and outer membrane and a layer of peptidoglycan in between. The outer leaflet of the outer membrane is primarily composed of lipopolysaccharides (LPS), but also contains outer membrane proteins (OMPs) (Sun et al, 2022). The bacterial cell envelope exhibits some degree of spatial heterogeneity, as peptidoglycan synthesis and OMP insertion primarily take place at the division poles (Mamou et al, 2022). Due to this process, older OMPs and peptidoglycan shift towards the old poles, where mature peptidoglycan inhibits OMP insertion (Mamou et al, 2022). Despite these spatial differences within the bacterial cell envelope, its impact on complement deposition and MAC-mediated killing has not yet been investigated.

Stable insertion of MAC pores into the bacterial outer membrane eventually results in inner membrane damage, which is the crucial event for bacterial cell death (Benn et al, 2024; Heesterbeek et al, 2019a, 2019b). Although it is hypothesized that inner membrane damage is an indirect effect of outer membrane damage, the critical event for MAC-mediated inner membrane damage remains unknown. Structural imaging studies revealed that MAC pores assembled from purified complement components C5b6 and C7-C9 form an asymmetric pore that is able to rupture single lipid bilayers of liposomes (Menny et al, 2018; Serna et al, 2016; Sharp et al, 2016). MAC-mediated killing of Gram-negative bacteria requires a more complicated process due to the complex structure of the bacterial cell envelope. In order to properly insert into the bacterial outer membrane and damage the bacterial inner membrane, MAC pores need to form close to the bacterial surface by convertase enzymes (Heesterbeek et al, 2019a). Atomic Force

---

[1]Department of Medical Microbiology, University Medical Center Utrecht, Utrecht University, Utrecht, The Netherlands. [2]Department of Medical Engineering and Applied Social Sciences, University of Applied Sciences Upper Austria, Linz, Austria. ✉E-mail: d.a.c.heesterbeek-2@umcutrecht.nl

Microscopy (AFM) on *E. coli* revealed that these bacteria can become fully covered with convertase-generated MAC pores at high complement concentrations and after sufficient incubation times (Benn et al, 2024). It is, however, unknown where MAC pores localize at the moment inner membrane damage is triggered, and whether some locations within the bacterial cell envelope are more vulnerable to MAC insertion.

Here, we visualized MAC localization on *E. coli* over time and reveal that initial MAC deposition occurs with a distinct localization pattern. By simultaneously capturing the moment of inner membrane damage, we show that MAC insertion at new bacterial poles correlates with bacterial inner membrane damage. These polar MAC pores also severely impair bacterial growth and division. Altogether, these findings shed light on how complement molecules attack the complex cell envelope of Gram-negative bacteria.

## Results

### MAC pores that trigger inner membrane damage are deposited with a distinct localization pattern

First, we wanted to investigate whether bactericidal MAC pores preferentially localize at certain parts of the bacterial surface. Therefore, we analyzed MAC deposition on *E. coli* MG1655 over time and captured the moment when the number of MAC pores was just sufficient to cause bacterial killing. To solely look at the effect of MAC on bacterial killing in the absence of other serum components, we pre-labeled bacteria with C5 convertase enzymes by incubating them with C5-depleted serum (Heesterbeek et al, 2019a). Then, after washing, convertase-labeled bacteria were incubated with purified MAC components (C5-C9) at a concentration of 2.5% serum equivalent (Muts et al, 2023) for different amounts of time (Fig. 1A). MAC deposition was visualized by using C9 that was fluorescently labeled with AF647. Previously, we showed that inner membrane damage is a critical event for MAC-mediated bacterial killing (Heesterbeek et al, 2019a, 2019b). Therefore, to measure at which timepoint MAC pores were able to induce bacterial cell death, we stained for inner membrane damage with a naturally membrane impermeable DNA dye (Sytox Blue).

By using flow cytometry, we first confirmed that MAC deposition increased gradually over time and that these pores could indeed induce inner membrane damage (Fig. 1B; Appendix Figs. S1A,B and S2A). To visualize where MAC pores localize at the moment inner membrane damage is triggered, we imaged the samples that were exposed to MAC components for different amounts of time by widefield fluorescence and phase-contrast microscopy. Since bacteria were in different stages of the cell cycle within one sample, we categorized them into two groups: 'rod-shaped' and 'dividing' (Fig. 1C). The efficiency of MAC deposition and inner membrane damage slightly varied between bacteria (Appendix Fig. S2B). Consistently among the samples however, MAC deposition initiated at the division poles on bacteria with a clear septum at mid-cell (dividing) (Fig. 1D). On bacteria that were just formed after cell separation (rod-shaped), MAC pores localized on only one of the two outer poles (Fig. 1D). On both groups of bacteria, polar MAC deposition was sufficient to cause inner

membrane damage. MAC deposition increased over time until bacteria were completely covered with MAC pores, consistent with previous studies (Benn et al, 2024). To confirm that this localization pattern is representative for other bacteria within the sample, we developed a high-throughput image analysis pipeline to identify individual bacteria within our images, categorize them into different growth stages (Spahn et al, 2022) and determine the average fluorescence intensity profile along the long axis of each cell (Appendix Fig. S3). The model was trained to distinguish dividing, rod-shaped, intermediate, defocused and clustered bacteria, excluding the last two categories from analysis. The growth stage classification (YOLO) model could identify each class with relatively high precision (Appendix Fig. S4). For the most important classes, dividing and rod-shaped, an additional filtering step based on bacterial length was added to further reduce misclassification. Furthermore, the model allowed us to select for bacteria with a damaged inner membrane by including a threshold based on the Sytox intensity. This advanced and innovative image analysis pipeline can be applied to large datasets and therefore allowed us to quantify average fluorescence distributions on multiple bacteria. To prove that the algorithm works correctly, it was tested on simulations of rod-shaped and dividing bacteria that were fully covered with fluorescent signal and showed random fluorescence distribution (Appendix Fig. S5). By applying the pipeline to a set of images of a timepoint at which we first start to observe bacteria with inner membrane damage (8-min timepoint), we demonstrate that the specific MAC localization shown in Fig. 1D is representative for other bacteria with a damaged inner membrane in the sample (Fig. 1E,F). By orienting bacteria based on the C9-AF647 signal, having the side with the highest fluorescence intensity on the left, we confirm that MAC pores preferentially localize at mid-cell on dividing bacteria, and on one of the outer poles of rod-shaped bacteria with a damaged inner membrane. To show that this localization pattern can be observed on more *E. coli* strains, we performed the same experiment on a MAC-sensitive clinical *E. coli* isolate, EC10. We found the same localization pattern of MAC pores on EC10, also correlating with bacterial inner membrane damage (Fig. EV1; Appendix Fig. S6).

Complement activation can occur through three different pathways: the classical pathway, triggered by antibody-antigen binding; the lectin pathway, triggered by mannose-binding lectin or ficolin binding; and the alternative pathway, triggered by spontaneous C3 hydrolysis, but also acting as an amplification loop (Dunkelberger and Song, 2010). Since pre-incubation of *E. coli* in C5-depleted serum could result in complement activation through each of these routes, we investigated whether only antibody-dependent complement activation triggers the same preferential MAC localization. Therefore, we performed a similar analysis in a fully purified complement assay in which only the classical pathway is activated. All complement components necessary for classical pathway activation were added, including C1-inhibitor to prevent non-specific fluid-phase activation of C1 (Fig. 2A) (Muts et al, 2023). To activate complement, we used an antibody that recognizes GlcNAc moieties present in the LPS of *E. coli* (Muts et al, 2025). At the antibody concentration that was used in our experimental setup, bacteria were fully covered with IgM without showing preferential localization (Fig. EV2). Upon exposure to MAC components, C9 deposition and inner membrane damage

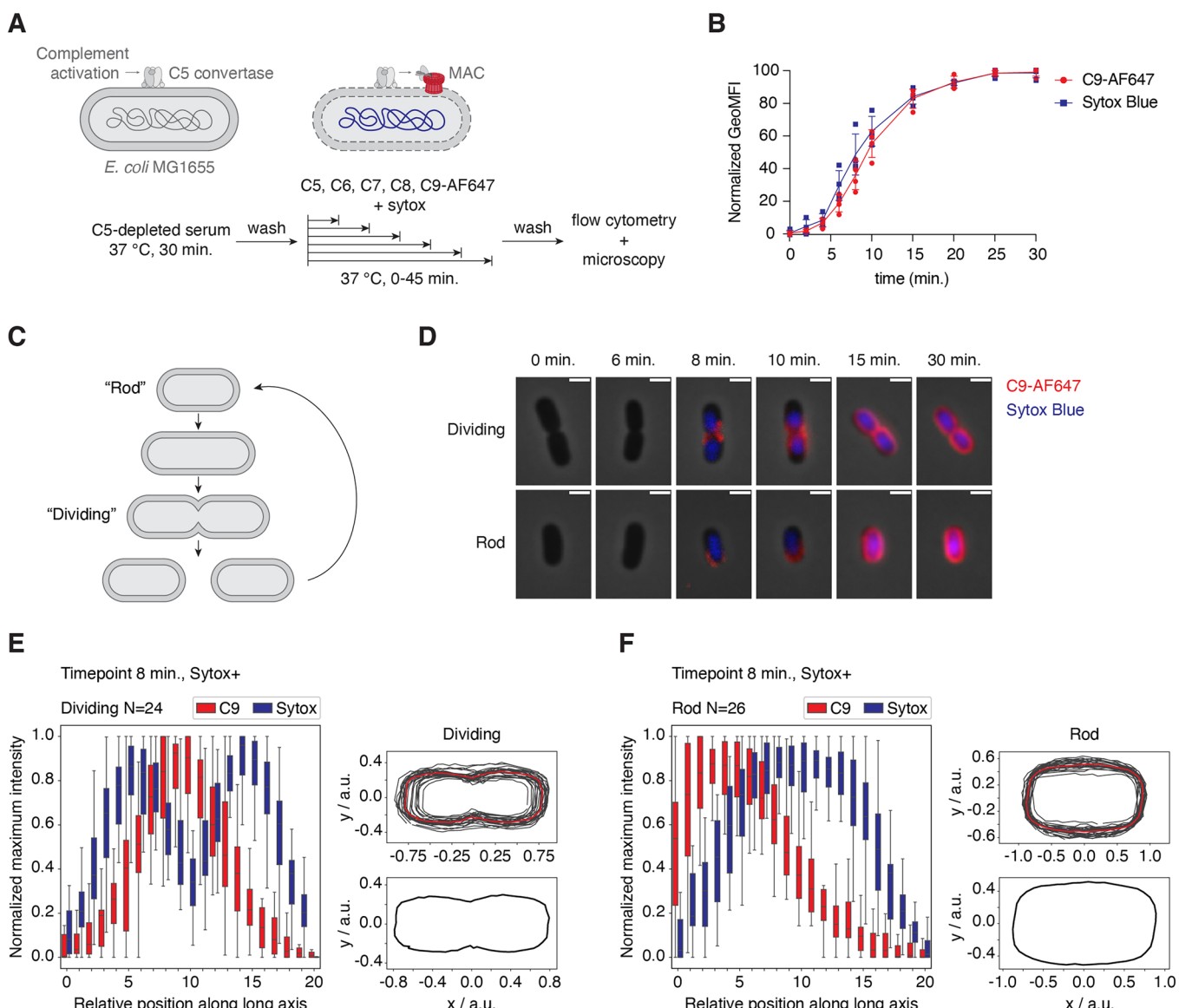

**Figure 1. MAC pores that trigger inner membrane damage are deposited with a distinct localization pattern.**

(A) Schematic overview of the experimental setup used to study MAC deposition and inner membrane damage in time. (B) C9-AF647 deposition and inner membrane damage (Sytox Blue) in time, measured by flow cytometry. (C) Schematic overview of the different stages that occur during bacterial growth and division. (D) Phase-contrast and overlayed fluorescence images of selected bacteria at different timepoints. C9-AF647 is shown in red and Sytox in blue. (E) On the left, normalized average C9 and Sytox distribution along the long axis of dividing bacteria after 8 min incubation with C5-C9. On the right, the average shape of bacteria that were classified as dividing. (F) On the left, normalized average C9 and Sytox distribution along the long axis of rod-shaped bacteria after 8 min incubation with C5-C9. On the right, the average shape of bacteria that were classified as rod-shaped. Data information: In (B), GeoMFI values were normalized by calculating the percentage of the maximum value after log-transformation. Data represent individual values with mean ± SD of four biological replicates. In (D), Individual bacteria have been chosen to reflect an average bacterium in the sample and are representative for three biological replicates. Scale bars: 2 μm. In (E, F), each box represents the interquartile range (IQR) of the data, with the center line indicating the median relative intensity. The whiskers extend to the most extreme values within 1.5× IQR. Data analysis was performed on all five images of one biological replicate, containing 24 dividing and 26 rod-shaped bacteria. Source data are available online for this figure.

increased gradually over time (Fig. 2B; Appendix Fig. S7A). In contrast to the serum setup, MAC deposition started earlier, and we observed a slight delay in inner membrane damage compared to the moment we first detected C9 (Figs. 1B and 2B). Despite these subtle differences, MAC also localized at the division poles on dividing bacteria and at one of the two outer poles on rod-shaped bacteria at the moment inner membrane damage was first induced (Fig. 2C,D; Appendix Fig. S7B). Together, these results show that

bactericidal MAC pores have a distinct localization pattern on *E. coli.*

## MAC pores preferentially deposit at the new pole of *E. coli*

Next, we investigated whether MAC pores preferentially localize at a specific pole on rod-shaped bacteria. Bacterial poles can be

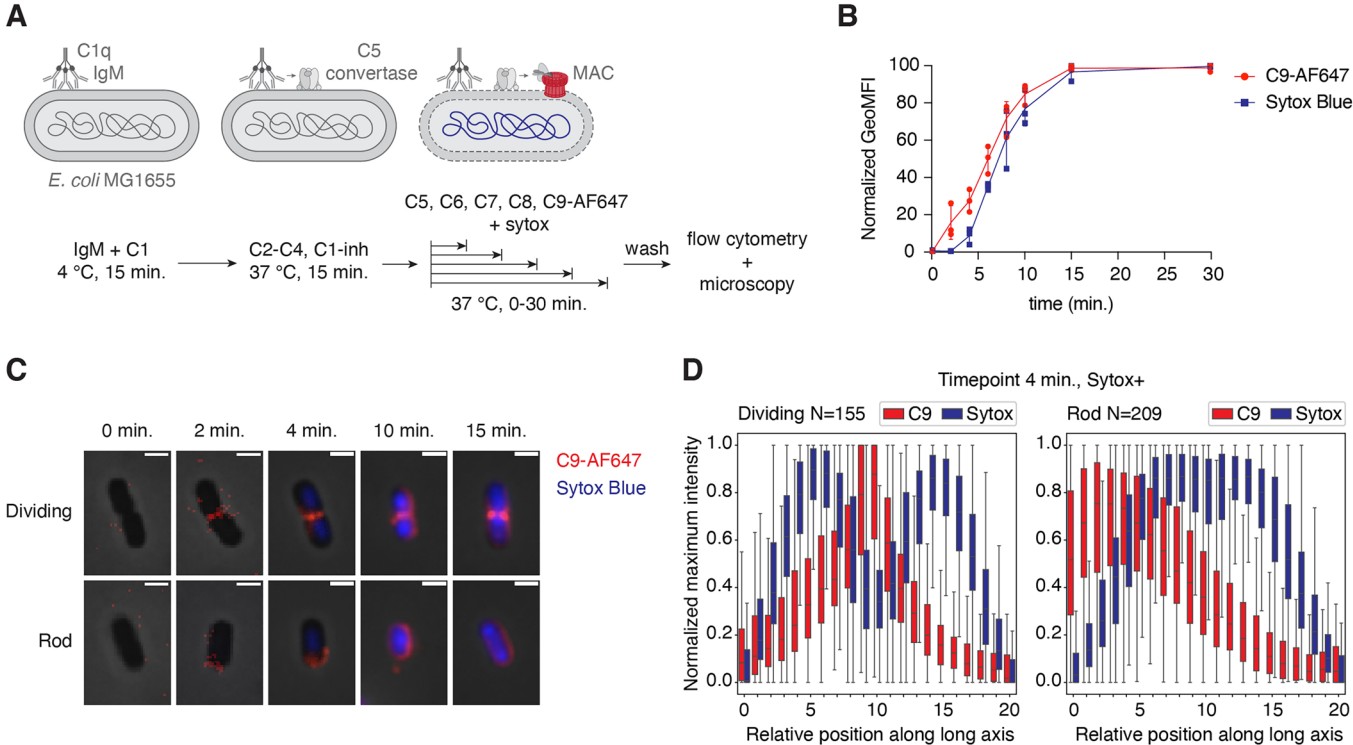

**Figure 2. Specific localization of MAC pores after antibody-mediated complement activation.**

(A) Schematic overview of the purified classical pathway setup used to study MAC localization and inner membrane damage in time. (B) C9-AF647 deposition and inner membrane damage (Sytox Blue) in time, measured by flow cytometry. (C) Phase-contrast and overlayed fluorescence images of selected bacteria at different timepoints. C9-AF647 is shown in red and Sytox in blue. (D) Normalized average C9 and Sytox distribution along the long axis of dividing and rod-shaped bacteria after 4 min incubation with C5-C9. Data information: In (B), GeoMFI values were normalized by calculating the percentage of the maximum value after log-transformation. Data represent individual values with mean ± SD of three biological replicates. In (C), individual bacteria have been chosen to reflect an average bacterium in the sample and are representative for three biological replicates. Scale bars: 2 μm. In (D), each box represents the interquartile range (IQR) of the data, with the center line indicating the median relative intensity. The whiskers extend to the most extreme values within 1.5× IQR. Data analysis was performed on all 19 images of one biological replicate, containing 155 dividing and 209 rod-shaped bacteria. Source data are available online for this figure.

classified as old or new, the latter being derived from the division pole in the previous round of division (Fig. 3A). Given that MAC deposits at the division poles of dividing bacteria, we hypothesized that it localizes at the new pole of rod-shaped bacteria. To distinguish the old and new poles, we used an approach to fluorescently label the old poles of *E. coli* with HCC-amino-D-alanine (HADA). HADA is a fluorescent D-alanine which is incorporated into newly synthesized peptidoglycan. By first adding HADA to the growth medium of *E. coli* for 1 h, we ensured that the peptidoglycan became uniformly labeled (Appendix Fig. S8). After washing, we allowed the bacteria to grow in the absence of HADA for another 45 min. Since peptidoglycan is primarily synthesized at the bacterial septum during cell division, the formation of new, unlabeled peptidoglycan causes HADA-labeled peptidoglycan to move towards the old poles (Fig. 3A; Appendix Fig. S8 and (Kuru et al, 2012)). During this process, bacteria were simultaneously labeled with convertases and exposed to purified MAC components (Fig. 3B), similar to the setup described in Fig. 1A. By using this method, we show that MAC pores localize at the poles that are not labeled with HADA, indicating that they specifically localize at new bacterial poles (Figs. 3C and EV3A,B). Analysis of a dataset of >3900 individual bacteria was used to validate the localization of MAC pores at the new pole of rod-shaped *E. coli* (Fig. 3D). For

dividing bacteria, the HADA signal was mostly oriented towards the old pole of one of the two daughter cells, with MAC pores again localizing at the division poles (Fig. 3D). Together, these results confirm that MAC pores preferentially localize at the new poles of both dividing and rod-shaped bacteria.

The observation that MAC pores preferentially deposit at the new pole raises questions about the underlying mechanism. At some point in the bacterial cell cycle, preferential MAC deposition should switch from one of the outer poles to the middle of the cell. To capture this moment, we analyzed bacteria at an 'intermediate' growth stage, which are already elongated compared to rod-shaped bacteria but did not yet form a visible septum. Interestingly, we observed less MAC deposition on 'intermediate' bacteria, suggesting that MAC deposition is less efficient on bacteria in this growth stage (Fig. EV3C). To further investigate the mechanism underlying preferential MAC deposition at the new pole, we studied MAC localization on bacteria that were treated with antibiotics that inhibit different bacterial processes. We either inhibited protein synthesis with chloramphenicol (Mamou et al, 2022) or peptidoglycan synthesis with mecillinam (Kals et al, 2025) or cephalexin (Weiss et al, 1999). Mecillinam inhibits peptidoglycan synthesis along the lateral cell wall during cell elongation (PBP2 inhibitor), while cephalexin inhibits peptidoglycan synthesis at mid-cell,

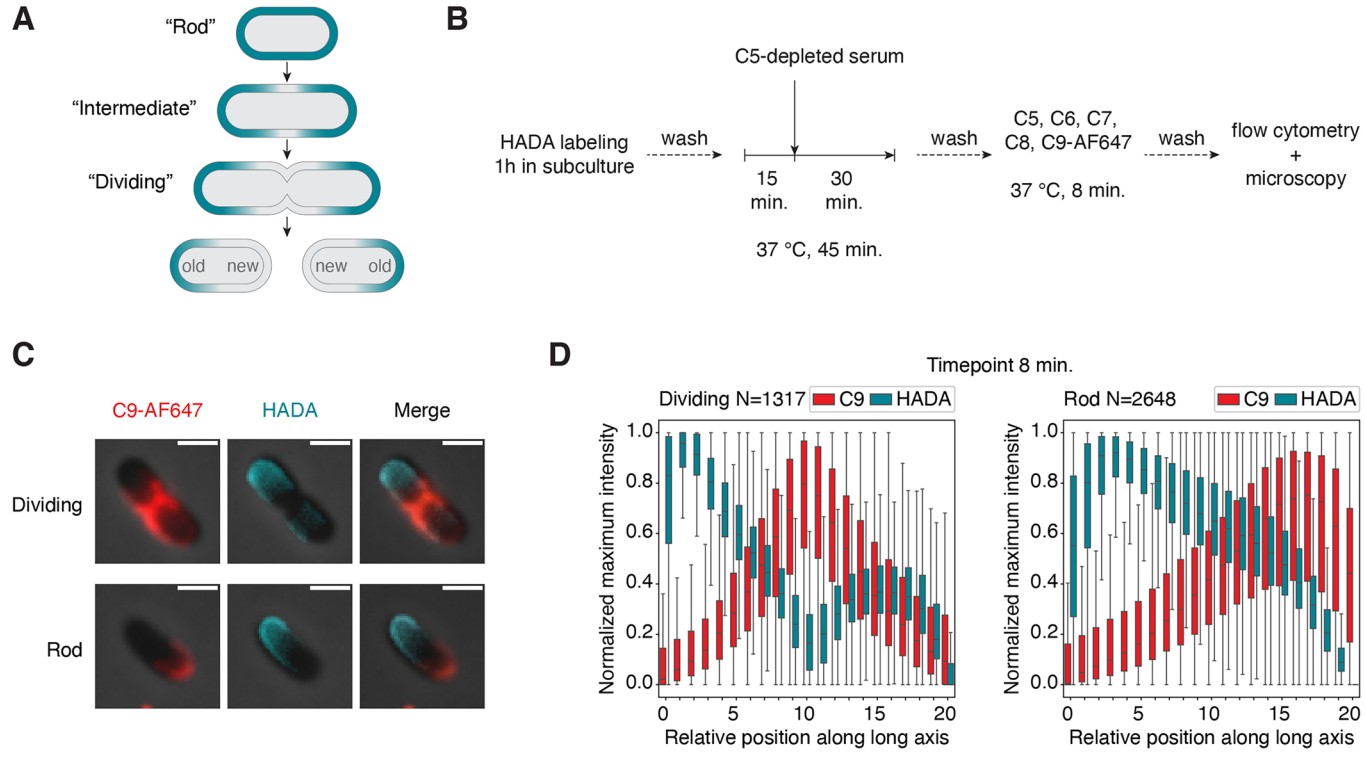

**Figure 3. MAC pores specifically localize at the new pole of *E. coli*.**

(A) Schematic representation of HADA distribution (in cyan) upon bacterial growth. After labeling bacterial peptidoglycan with HADA, new unlabeled peptidoglycan is synthesized at the bacterial septum, moving labeled peptidoglycan toward the old poles. (B) Schematic overview of the experimental setup used to label old bacterial poles with HADA and induce MAC deposition. (C) Phase-contrast and overlayed fluorescence images of selected bacteria after performing the assay as shown in (B). C9-AF647 is shown in red and HADA in cyan. (D) Normalized average C9-AF647 and HADA distribution along the long axis of dividing and rod-shaped bacteria. Data information: In (C), individual bacteria have been chosen to reflect an average bacterium in the sample and are representative for three biological replicates. Scale bars: 2 μm. In (D), each box represents the interquartile range (IQR) of the data, with the center line indicating the median relative intensity. The whiskers extend to the most extreme values within 1.5× IQR. Data analysis was performed on all 50 images of one biological replicate, containing 1317 dividing and 2648 rod-shaped bacteria.

which is required for septum formation (PBP3 inhibitor). Despite the altered shapes of the treated bacteria, we still observed localization of MAC pores on the new poles or, on cephalexin-treated bacteria, at distinct patches along the cell cylinder (Fig. EV4A). Although the localization pattern was similar, inhibition of protein synthesis resulted in less C9 deposition per bacterium. This indicates that active protein synthesis increases the efficiency of MAC deposition, but active division is not required to trigger preferential MAC deposition at the new pole.

## MAC pores at the new pole of *E. coli* impair bacterial cell division

Next, we analyzed how bacterial growth is affected by the deposition of MAC pores at the new pole. We have previously shown that the intensity of sytox influx correlates with a reduction of colony forming units (CFUs) (Heesterbeek et al, 2019a, 2019b). However, CFU counts only provide information about which percentage of a large population of bacteria survives and do not show any information about the behavior of individual bacteria. To study how MAC localization affects bacterial growth and division of individual bacteria, we used time-lapse microscopy to follow bacteria for 2.5 h after exposure to MAC components. First, we

confirmed that also in this experimental setup (Fig. 1A), an increase in inner membrane damage correlated with a decrease in CFUs (Fig. 4A). In line with the partial survival of bacteria after 6, 8 and 10 min of exposure to MAC components, the number of cell divisions per bacterium differed vastly within one sample (Fig. 4B; Movies EV1–4). To study whether the number of divisions was dependent on the amount of MAC pores per bacterium, we manually categorized bacteria based on inner membrane damage (Sytox-positivity) and the level of MAC deposition (Appendix Fig. S9). Bacteria were first classified as either Sytox-negative or -positive. Sytox-positive bacteria were further categorized into having polar C9 or being fully covered with C9. This categorization revealed that differences in bacterial growth could largely be explained by inner membrane damage, as bacteria with inner membrane damage showed drastically impaired cell division (Fig. 4C). Furthermore, most bacteria with MAC pores at the new pole were unable to grow and divide (Fig. 4C,D). Interestingly, in some cases, bacteria with inner membrane damage and polar MAC pores were able to partially escape killing (Fig. 4D). Analysis of the C9 and Sytox intensities on bacteria that had been incubated with MAC components for 6 min revealed a moderate negative correlation between the number of divisions and the fluorescence intensity of both C9 and Sytox (Appendix Fig. S10). This suggests

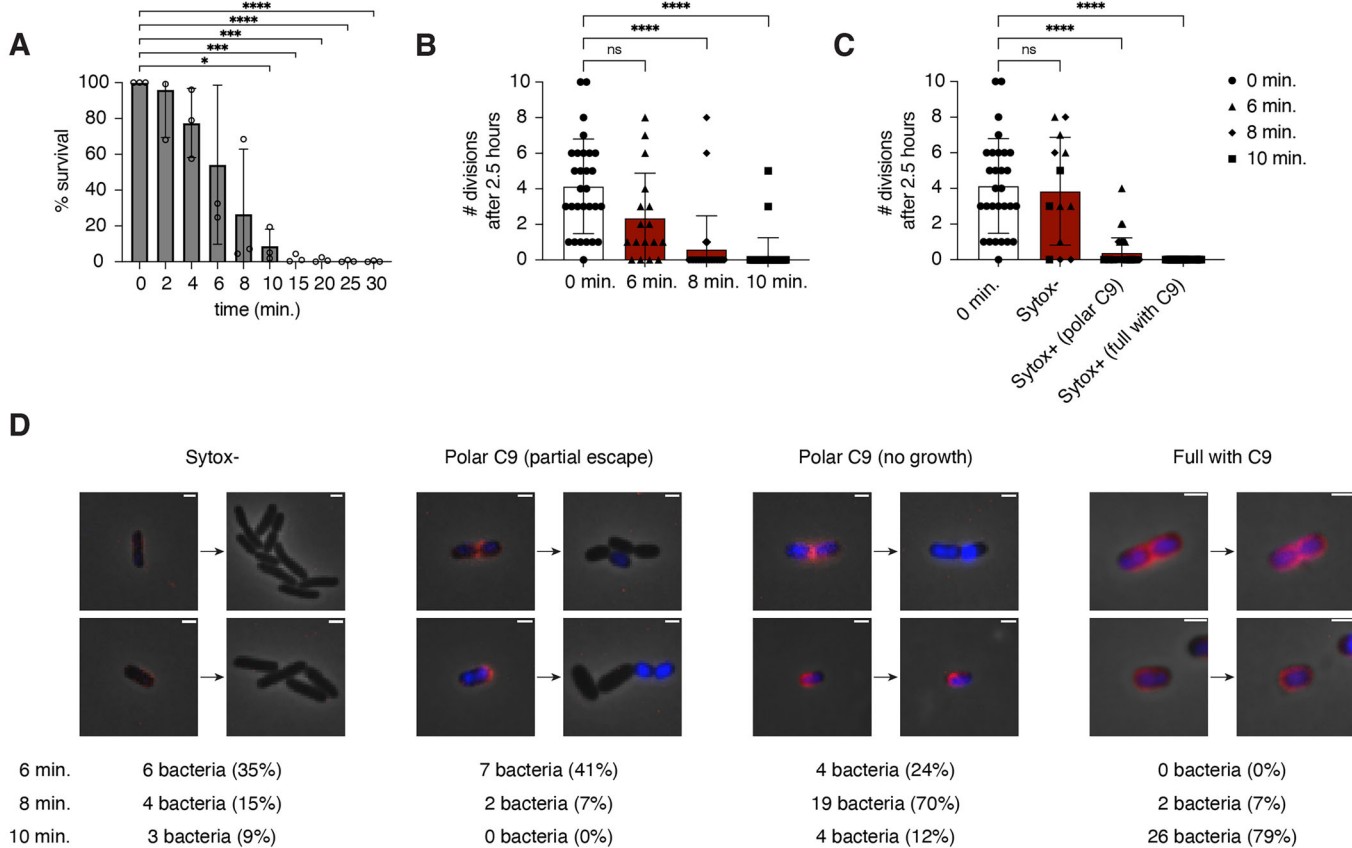

**Figure 4. MAC pores at the new pole of *E. coli* drastically impair bacterial cell division.**

(A) Bacterial survival after performing the assay as shown in Fig. 1A, based on CFU/ml. Percentage survival was calculated by normalizing to the 0-min timepoint, in which bacteria had not been exposed to MAC components. (B) The number of divisions of individual bacteria within 2.5 h, imaged after exposure to MAC components for the indicated amounts of time. (C) Graph as in (B), grouping bacteria exposed to MAC components based on Sytox-positivity and the localization of C9-AF647. (D) Images of bacteria that had been incubated with C5-C9 for 6–10 min, before and after allowing growth for 2.5 h. Images show examples of the groups shown in (C). Below are the number and percentage of bacteria showing each phenotype after 6, 8 or 10 min incubation with C5-C9. Data information: In (A), data represent individual values with mean ± SD of three biological replicates. Statistics was performed using a repeated measures one-way ANOVA (*P* = 0.0335) followed by Dunnett's post-hoc test, comparing each group with the 0-min timepoint. Only significant differences are shown. *P* < 0.05, ****P* < 0.001, *****P* < 0.0001. In (B, C), each data point represents the number of divisions made by an individual bacterium. Bars represent mean ± SD. Statistics was performed using a Kruskal–Wallis test (*P* < 0.0001) followed by a Dunn's post-hoc test, comparing each group with the 0-min timepoint. *P* < 0.05, *****P* < 0.0001. In (D), scale bars: 2 μm. Source data are available online for this figure.

that some bacteria can survive a low level of MAC deposition and inner membrane damage. Bacteria with an intact inner membrane at the start of the video were able to go through an average of ~4 divisions (Fig. 4C). There was a large variation in the number of divisions per bacterium, but this was also seen for bacteria that had not been exposed to MAC pores (0-min timepoint) (Fig. 4C). When bacteria were exposed to MAC components for 10 min, almost all imaged bacteria were fully covered with MAC pores and unable to divide (Fig. 4C,D). This correlates with almost no bacterial survival (CFUs) after 10 min of exposure to MAC components (Fig. 4A). Taken together, this shows that MAC pores at the new pole of *E. coli* severely impair bacterial cell growth and division.

## Deposition of MAC component C8 also starts at the new pole of *E. coli*

Next, we questioned what step in the complement cascade determines the preferential localization of C9 at the new pole. To study whether the C5b-8 complex also shows a distinct localization pattern or whether C9 localization may be caused by more efficient C9 polymerization at the new pole, we simultaneously measured C8 and C9 deposition. To do this, we exposed *E. coli* to MAC pores for 8 min using the same setup as described in Fig. 1 but using recombinant C8 that was directly labeled with AF488. At this timepoint, we observed co-localization between C8-AF488 and C9-AF647 at the bacterial (division) poles (Fig. 5A,B). Similarly, the first MAC component C5 also localized at the new pole at the moment of inner membrane damage in the presence of MAC components C6-C9 (Fig. EV5). We then wondered if C9, with its membrane insertion domain, may cause more efficient anchoring of early MAC components C5b-8 at the new pole. Therefore, we studied the deposition of C8 molecules, which possess their own membrane insertion domain, in the absence of C9. By exposing convertase-labeled bacteria to C5-C7 and C8-AF488 for 8 min, we show that C8 preferentially inserts at the bacterial poles independently of C9 (Fig. 5C,D). Finally, we investigated whether

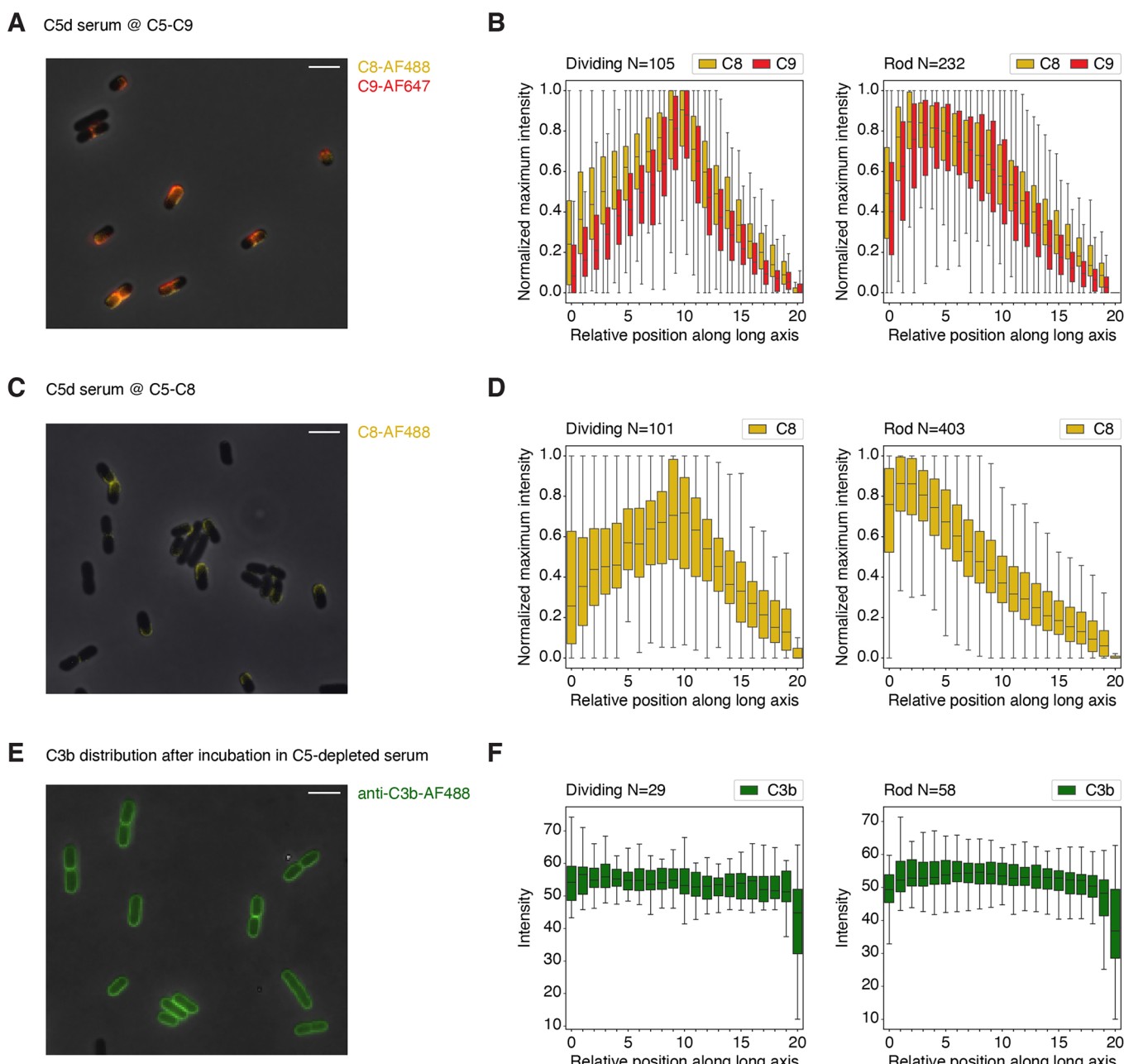

**Figure 5. Deposition of MAC component C8 also starts at the new pole of *E. coli*.**

(A) Convertase-labeled *E. coli* bacteria were exposed to C5-C7, C8-AF488 and C9-AF647 for 8 min. Localization of C8-AF488 (yellow) and C9-AF647 (red) was determined by phase-contrast and widefield fluorescence microscopy. (B) Data analysis of the sample shown in (A). Graphs show the normalized average C8-AF488 and C9-AF647 distribution along the long axis of dividing and rod-shaped bacteria. (C) Complement deposition on *E. coli* was induced as in (A), but in the absence of C9. Localization of C8-AF488 (yellow) was determined by phase-contrast and widefield fluorescence microscopy. (D) Data analysis of the sample shown in (C). Graphs show the normalized average C8-AF488 distribution along the long axis of dividing and rod-shaped bacteria. (E) Bacteria were incubated with 10% C5-depleted serum followed by staining for C3b using a fluorescently labeled monoclonal antibody. (F) Data analysis of the sample shown in (E). Graphs show the average anti-C3b-AF488 distribution along the long axis of dividing and rod-shaped bacteria. Data information: In (A, C), @ indicates a washing step. In (B, D, F), each box represents the interquartile range (IQR) of the data, with the center line indicating the median relative intensity. The whiskers extend to the most extreme values within 1.5× IQR. In (B), data analysis was performed on all 20 images of one biological replicate, containing 105 dividing and 232 rod-shaped bacteria. In (D), data analysis was performed on all 20 images of one biological replicate, containing 101 dividing and 403 rod-shaped bacteria. In (F), data analysis was performed on all three images of one biological replicate, containing 29 dividing and 58 rod-shaped bacteria. All images and graphs are representative for three biological replicates. Scale bars: 5 µm.

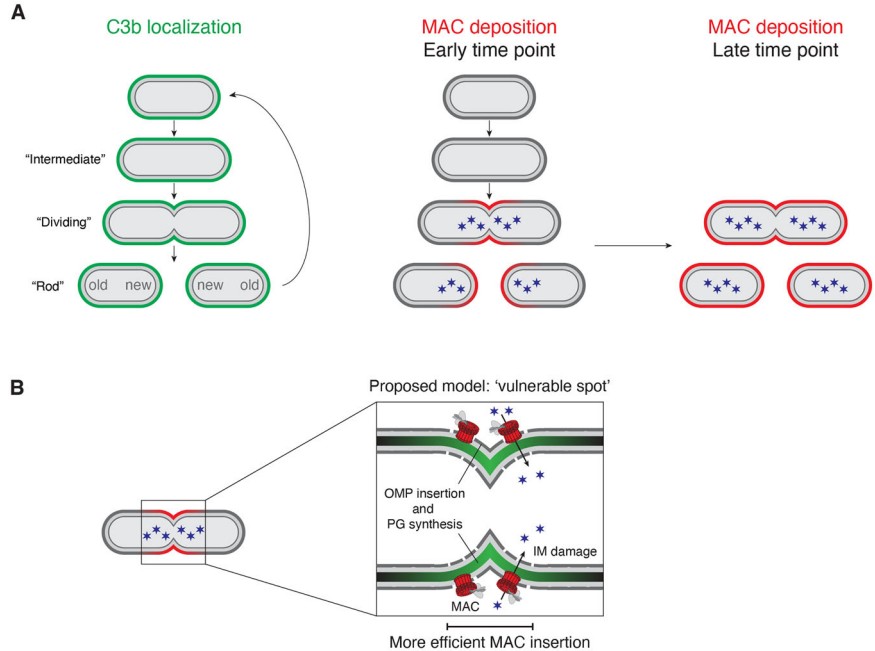

**Figure 6. Comprehensive model for MAC deposition on MAC-sensitive *E. coli*.**

(A) While complement is activated along the entire bacterial surface, resulting in homogeneous distribution of C3b, MAC is preferentially deposited at the new pole of *E. coli* (early time point). Over time, MAC deposition spreads out until bacteria become fully covered with MAC pores (late time point). (B) Proposed model for how the coordinated biogenesis of outer membrane proteins (OMPs) and new peptidoglycan (PG) at the division pole of *E. coli* could create a vulnerable spot at which more efficient insertion of bactericidal MAC pores takes place. Data information: Blue stars represent the influx of a membrane impermeable DNA dye due to inner membrane (IM) damage induced by MAC pores.

more efficient complement activation at the new pole could explain the localized insertion of MAC pores. Therefore, we measured the distribution of C3b molecules as a readout for complement activation. C3b molecules were evenly distributed over the bacterial surface after incubation in C5-depleted serum (Fig. 5E,F), indicating that the preferred polar insertion of C8 and C9 is not caused by localized C3b deposition on the bacterial surface. Altogether, this suggests that the preferential localization of MAC pores at the new pole of *E. coli* is determined by one of the initial steps of MAC formation.

## Discussion

In this study, we reveal that bactericidal MAC pores show a distinct localization pattern on *E. coli*. By capturing the start of MAC deposition, we show that MAC pores are first deposited at new bacterial poles, where they trigger inner membrane damage and halt bacterial cell division (Fig. 6). Whereas MAC pores initially insert at the new pole, we found that bacteria become fully covered with MAC pores within ~15 min of exposure to MAC components. This is in line with a recent study in which MAC-treated *E. coli* was imaged by AFM, which showed random distribution of MAC pores after these exposure times (Benn et al, 2024). Since bacterial cell death is triggered quickly after initial MAC deposition at the new pole, the question remains how relevant later-formed pores at different locations are in bacterial killing. We previously showed that a limited number of C8 molecules (roughly between 100 and

1000 per bacterium), and thus MAC pores, are sufficient to cause inner membrane damage (Heesterbeek et al, 2019a). Although bacterial killing correlated with polar MAC localization, we cannot exclude that non-polar pores contribute to bacterial killing as well. Future studies should point out whether MAC insertion at the new pole is a requirement for bacterial cell death, or whether MAC pores at other locations can also efficiently induce bacterial killing. If MAC pores at other locations can also induce inner membrane damage, the here-observed preferential localization of MAC pores is a matter of insertion efficiency at these places. A high local density of MAC pores at the new pole resulting from more efficient insertion may contribute to causing local large-scale disruption of the outer membrane.

In order to understand how the complement system attacks bacterial cells, it is essential to consider the impact of the complex structural organization of the bacterial cell envelope. Spatial differences within the structure of the outer membrane or underlying peptidoglycan layer could explain the mechanism underlying MAC deposition at new bacterial poles. Interestingly, the pattern of MAC localization seems to resemble images of the coordinated assembly of peptidoglycan and insertion of OMPs into the outer membrane of *E. coli* (Mamou et al, 2022). It was shown that upon cell division, bacteria preferentially insert OMPs at the division poles, depending on the synthesis of immature peptidoglycan at the same locations. Preferential insertion of MAC pores at these exact spots could indicate that the assembly of cell envelope components creates a vulnerable spot within the cell envelope during bacterial division. In addition, differences in lipid

composition of the bacterial outer membrane may also play a role in preferential MAC deposition at the new pole. The major component of the Gram-negative outer membrane, LPS, is evenly distributed across the bacterial membrane and, in contrast to the mid-cell biased insertion of OMPs, inserted uniformly across the outer membrane (Kumar et al, 2025). Whereas another lipid, cardiolipin, is primarily present at the bacterial poles, this localization is not specific for the new pole (Renner and Weibel, 2011). Therefore, differential localization of LPS or cardiolipin is unlikely to determine the initiation of MAC insertion at the new poles. Local differences in densities of lipids, however, might create sites for preferred MAC insertion.

The observation that MAC deposition initiates at new bacterial poles, which are formed during cell division, poses the question whether active division is a requirement for MAC-dependent killing of Gram-negative bacteria. Inhibition of peptidoglycan synthesis along the lateral cell wall with mecillinam did not influence the insertion efficiency or the localization pattern of MAC pores. When peptidoglycan synthesis was inhibited with cephalexin, preventing septum formation, MAC deposition was most efficient at mid-cell while also being detected along the cell cylinder. This suggests that actual septum formation is not the main determinant for preferential MAC deposition to occur. However, the increased vulnerability at mid-cell may still arise from cell division processes that took place before exposure to the antibiotics. Although inhibition of protein synthesis with chloramphenicol did not change the localization pattern of initial MAC deposition, we did observe MAC deposition on only a small proportion of the chloramphenicol treated bacteria. This shows that MAC deposition is less efficient on bacteria in which protein synthesis is inhibited. The fact that we observed less MAC deposition on bacteria in an 'intermediate' growth stage suggests that bacteria in the elongation phase are less vulnerable to MAC insertion. Visualizing the localization of cell division machinery and OMP insertion in bacteria at different growth stages or after antibiotic treatment may give new insights into what creates a vulnerable spot that allows for more efficient MAC insertion.

The here-described vulnerability of the new pole may not be unique to the insertion of MAC pores but may also apply to other membrane-attacking compounds. For example, the antimicrobial peptide Cecropin A was shown to preferentially attack septating cells and cause leakage of periplasmic GFP initially from the division poles (Rangarajan et al, 2013). This supports the hypothesis that bacterial cell division may coincide with increased vulnerability to compounds that attack the bacterial membrane.

Time-lapse microscopy of bacterial growth after MAC deposition demonstrated that polar MAC deposition and inner membrane damage correlate with bacterial cell death. Due to heterogeneity in the exact timing of MAC deposition, some bacteria within a sample are already killed while others are not yet. When categorizing bacteria based on the localization of the MAC, we found that MAC deposition at new bacterial poles was sufficient to cause inner membrane damage and halt bacterial growth. This again proves that relatively low amounts of MAC pores are sufficient to induce bacterial killing. Interestingly, in some cases we noticed that bacteria with limited MAC deposition but detectable inner membrane damage were able to survive and divide again. This partial escape of bacterial killing, which also caused a loss of Sytox signal over time, suggests that bacteria can repair low degrees of

inner membrane damage. However, MAC deposition at the bacterial poles was often sufficient to stop bacterial growth entirely.

The exact step within the complement cascade that causes preferred polar MAC deposition remains to be determined. We show that MAC component C8 has the same localization pattern as C9, indicating that C8 is able to efficiently insert into the new pole with its membrane insertion domain, independent of C9. The anchoring and polymerization of C9 may however also still be more efficient at the new pole, which would subsequently also lead to a higher number of fully polymerized, and therefore functional pores at these locations. We observed homogeneous distribution of the IgM antibody targeting GlcNAc moieties present in the LPS core of *E. coli*. This indicates that this antibody does not preferentially localize at certain locations, which would have caused more efficient complement activation at these spots. This is further supported by the random distribution of C3b, again suggesting that polar MAC localization is not caused by more potent complement activation at the new pole. While more efficient insertion of MAC precursors into potentially weak bacterial poles seems a likely explanation for the preferred polar MAC deposition, it could also be caused by enhanced activity of C5 convertases at these spots. For instance, even though C3b seems to be uniformly distributed over the bacterial surface, there might be a subtle difference in the local density of C3b at the new bacterial pole. Since high densities of C3b are essential to form C5 convertase enzymes, which trigger MAC formation by cleaving C5, subtle differences in C3b may result in more efficient C5 cleavage at the new poles. Follow-up studies are needed to study whether preferential MAC localization at the new poles may be caused by more efficient C5 conversion at these locations.

Our observation of polar MAC localization on the MAC-sensitive *E. coli* MG1655 strain was validated on a MAC-sensitive *E. coli* clinical isolate. Future studies are needed to investigate whether these findings can also be translated to other Gram-negative bacterial species. To our knowledge, specific localization of MAC pores on Gram-negative bacteria has not yet been described before. In contrast, the localization of MAC pores has previously been tested on four different Gram-positive bacterial species and varied from being predominantly present at the division pole on one bacterial strain (*S. pyogenes*) to a more random distribution (Berends et al, 2013). However, as these bacteria do not have an outer membrane in which MAC pores can insert, it does not describe the localization of lethal MAC pores that induce inner membrane damage. Although Gram-positive bacteria possess a vastly different cell envelope and are intrinsically MAC-resistant, it may be interesting to include these and other Gram-negative species with different sensitivities to MAC-dependent killing in follow-up studies. Extending our current analysis to a larger bacterial panel may guide us towards a better understanding of what determines MAC localization and how this relates to MAC sensitivity. A relatively large proportion of pathogenic bacteria is resistant to MAC-mediated killing, which is often linked to improper membrane insertion of MAC pores (Doorduijn et al, 2022). On Gram-negative bacteria, the insertion of MAC pores is likely influenced by the presence of an O-antigen (Burns and Hull, 1998; Doorduijn et al, 2021) or polysaccharide capsule (Leying et al, 1990; Suerbaum et al, 1994). Since the laboratory *E. coli* strain used in this study lacks an O-antigen and capsule, testing MAC localization on Gram-negative bacteria with varying O-antigen

lengths and capsule thicknesses may reveal whether these structures influence the distribution of MAC pores on the bacterial surface. If MAC insertion at the new pole turns out to be required for MAC-mediated killing, bacteria might have acquired defense mechanisms against MAC-dependent killing by preventing MAC insertion at this location.

Taken together, this study provides novel insights into how Gram-negative bacteria are killed by MAC pores. The growing problems with antibiotic resistant bacteria (Naghavi et al, 2024) have increased the urgency to invest in the development of alternative antimicrobial therapies (Keck et al, 2024; Kumar et al, 2021; Sceglovs et al, 2025). These for example include the use of antibodies to activate the complement system and trigger MAC formation, the discovery of new antimicrobial agents that directly act on Gram-negative bacteria or the use of lytic bacteriophages. The here-presented fundamental insights into what locations of the bacterial cell envelope may be most vulnerable to components that act on the bacterial outer membrane could provide valuable information in the design of these new antibacterial therapies.

# Methods

### Reagents and tools table

| Reagent/resource | Reference or source | Identifier or catalog number |
|---|---|---|
| **Experimental models** | | |
| *E. coli* strain MG1655 | Derived from Escherichia coli str. K12 | NCBI:txid511145 |
| *E. coli* clinical isolate EC10 | Patient derived - According to European/Dutch procedures that are documented within the local setting of our hospital (UMC Utrecht). No data from the volunteers was collected. | |
| **Recombinant DNA** | | |
| pUPE vector expressing C2 | Gift from U-Protein Express BV | |
| pcDNA3.4-MCS | Struijf et al, 2023 | |
| pRSET-B | Thermo Fisher Scientific | |
| **Antibodies** | | |
| Human-anti-GlcNAc IgM | Produced in-house (Muts et al, 2025) | Clone 4497 |
| Goat F(ab')2 anti-kappa-AF647 | Southern Biotech | 2062-31 |
| Mouse anti-C3b-AF488 | Produced in-house. Production details are specified in "Methods" of the manuscript. Hybridoma cells producing MoAb bH6 to C3b were kindly provided by Peter Garred (University of Copenhagen) and Tom Eirik Mollnes (University of Oslo). | Clone bH6 |
| Goat Anti-Human C5 | Complement Technology | A220 |
| Donkey Anti-Goat IgG | Jackson ImmunoResearch | 705-606-147 |
| **Oligonucleotides and other sequence-based reagents** | | |
| **Chemicals, enzymes and other reagents** | | |
| Complement protein C1 | Complement Technology | A098 |
| Complement protein C2 | This study | |
| Complement protein C3 | Isolated from human plasma (Lambris et al, 1980) | |
| Complement protein C4 | Complement Technology | A105 |
| Complement protein C5 | This study | |
| Complement protein C6 | This study | |
| Complement protein C7 | This study | |
| Complement protein C8 | Complement Technology | A125 |
| Complement protein C8-AF499 | This study | |
| Complement protein C9 | This study | |
| Complement protein C9-AF647 | This study | |
| C1-inhibitor | Complement Technology | A140 |
| C5-depleted serum | Complement Technology | A320 |
| HIS-TEV-Sortase | This study | |
| AF488 NHS Ester | Thermo Fisher Scientific | A20000 |
| GGG-AF488 | Thermo Fisher Scientific | A6198 |
| GGG-AF647 | Click Chemistry Tools | 1556 |
| Sytox Blue | Thermo Fisher Scientific | S11348 |
| HADA | Biosynth | DQD73310 |
| Chloramphenicol | Sigma-Aldrich | C0378 |
| Cephalexin | Melford | C59000 |
| Mecillinam | TargetMol | T21369 |
| **Software** | | |
| Python | | Version 3.11.9 |
| FlowJo | | V10 |
| GraphPad Prism | | Version 9.4.1. |
| FIJI | | Version 2.14.0/ 1.54 f |
| **Other** | | |
| SeaPlaque agarose | Lonza | 50101 |
| 22 mm × 22 mm × 0.96 mm siliconized glass cover slides | Hampton Research | HR3-233 |
| Microscope slides | Epredia | 15545650 |
| 22 mm × 22 mm coverslips | Superior Marienfeld | |
| EXPI293F cells | Thermo Fisher | |

## Bacterial strains

Unless specified in the figures, experiments were performed on *E. coli* strain MG1655. The *E. coli* clinical isolate (EC10) was isolated from a patient with bacteremia and obtained from the Medical Microbiology Department of the University Medical Center Utrecht. For each experiment, a single bacterial colony was picked

from a blood agar plate and grown overnight in LB medium, shaking at 37 °C. The next day, cultures were diluted 1:100 in LB medium and grown until mid-log phase ($OD_{600\,nm}$ = 0.35–0.65). When peptidoglycan was labeled with HCC-amino-D-alanine (HADA), 0.1 mM was added to the subculture during the last hour of growth. Unless stated otherwise, incubations were performed in RPMI 1640 medium supplemented with 0.05% human serum albumin (RPMI/HSA). Washing steps were performed by spinning down bacteria for 2 min at 10,000×g.

## Serum and complement proteins

C5-depleted serum and complement proteins C1, C4, C8 and C1-inhibitor were obtained from Complement Technology. While commercial C8 was used for experiments with unlabeled C8, we later produced fluorescently labeled C8 in-house. His-tagged C2, C5, C6, C7, C8 and C9 were expressed in Expi293F cells (Thermo Fisher) and purified as described below. Fluorescent labeling of C8-AF488 and C9-AF647 is also described below. C3 was isolated from human plasma as described previously (Lambris et al, 1980). Monoclonal anti-GlcNAc (4497) IgM was produced as described previously (Muts et al, 2025). Sytox Blue was purchased from Thermo Fisher. Concentrations of complement components in 100% serum: 135 μg/ml C1, 20 μg/ml C2, 1250 μg/ml C3, 400 μg/ml C4, 70 μg/ml C5, 64 μg/ml C6, 56 μg/ml C7, 55 μg/ml C8, 60 μg/ml C9, 180 μg/ml C1-inhibitor.

## Production of complement components C2, C6, C7, C8 and C9

Complement component C2 was expressed with a N-terminal HIS-TEV site and 3 alanines at its C-terminus (GS-HHHHHH-DYDIPSS-ENLYFQG-C6-AAA). The pUPE vector, expressing C2, was a gift from U-Protein Express BV. C5, C6, C7, and C9 (NM_001737;AA22-559) were cloned in pcDNA3.4-MCS, as described before (Struijf et al, 2023). C5 and C6 were cloned to express a C-terminal 1xLinker (1xGGGGS)-LPETGG-6xHIS tag, whereas C7 and C9 contained a C-terminal (3xGGGGS)-LPETGG-6xHIS tag. To express C8, the genes for C8α (NM_000562), C8β (NM_000066) and C8γ (NM_000606) were cloned in 3 separate pcDNA3.4-MCS vectors. The cystatin(S) signal peptide was used for all three genes. C8α (AA31-584) was cloned to express a C-terminal (3xGGGGS)-LPETGG-StrepTagII (SAWSHPQFEK). Both C8β (AA55-591) and C8γ (AA21-202) were cloned to express a C-terminal (3xGGGGS)-LPETGG-6xHIS tag. The LPETG sequence in the tag of each C8 subunit was used for later triple labeling of C8 via sortagging (Popp et al, 2007). HIS-tagged C2, -C5, -C6, -C7, and -C9 proteins were expressed in EXPI293F cells, as before (Struijf et al, 2023). In short, EXPI293F cells were transfected with 0.5 μg DNA/ml cells, containing 50% empty, "dummy", vector using a ratio of 5:1 PEI/DNA (w/w). After 4 h, 1 mM valproic acid was added to the transfected cells and cell supernatants were collected after 5 days of expression. Supernatants were dialyzed against 50 mM Tris, 500 mM NaCl; pH 8.0 and HIS-tagged C2, -C5, -C6, -C7, and -C9 proteins were isolated on HiTrap Chelating HP columns (Cytiva), according to the manufacturer's description. Expression of StrepTagII-tagged C8 in EXPI293F cells was performed as described above, using a total of 0.5 μg DNA/ml cells with a ratio of C8α:C8β:C8γ:Empty vector of

0.24:0.33:0.24:0.19 per μg total used plasmid. After 5 days of expression, EXPI293F cell supernatant was buffer exchanged to 100 mM Tris, 150 mM NaCl, 1 mM EDTA; pH 8.0 using the QuixStand benchtop system (GE HealthCare). To inactivate residual biotin, 1.8 μl/ml Biolock (IBA Lifesciences) was added and incubated for 30 min. Then, the C8 cell supernatant was applied to a 1 ml Strep-Tactin®XT 4Flow® FPLC column (IBA Lifesciences) and C8 was eluted according to the manufacturer's description.

## Fluorescent labeling of complement components C8 and C9

HIS-TEV-Sortase 7+ (Jeong et al, 2017) was cloned in the pRSET-B vector (Thermo Fisher Scientific) and expressed in BL21(DE3) pLysS E. coli (Thermo Fisher Scientific) after induction with 1 mM Isopropyl β-D-1-iogalactopyranoside (IPTG) for 4 h. HIS-TEV-Sortase was isolated from a HiTrap chelating HP column under native conditions and eluted using an imidazole gradient.

For triple labeling of C8, 4.5 μM C8 was treated for 2 h at 4 °C with 45 μM ggg-AF488 (Thermo Fisher Scientific) and 0.9 μM HIS-TEV-Sortase 7+ in sortase buffer (10 mM $CaCl_2$, 50 mM Tris, 150 mM NaCl; pH 8.0). Then, 0.1 volume of 10× Strep-Tactin buffer (500 mM Tris, 150 mM NaCl, 110 mM EDTA; pH 8.0) was added, before application to a 1 ml Strep-Tactin®XT 4Flow® FPLC column (IBA Lifesciences). The flow through was collected, concentrated to 500 μl, and subsequently applied to a Superdex 200 increase 10/300 GL (Cytiva) gel filtration column. Fractions containing AF488-labeled C8 were collected and had a AF488 dye to protein label (DOL) of 250%. To fluorescently label C9, 50 μM C9-3xGGGGS-LPETG-6xHIS was incubated for 2 h at 4 °C with 500 μM ggg-AF647 (Click Chemistry Tools) and 10 μM His-TEV-Sortase 7+ in sortase buffer. Then, 0.1 volume of 10× binding buffer (50 mM Tris, 3.5 M NaCl, 200 mM imidazole; pH 8.0) was added. To capture the HIS-TEV-Sortase and non-AF647 labeled (HIS-tagged) C9, the sample was loaded onto a HiTrap Chelating HP column. The flow through (C9-AF647) was collected, concentrated to 500 μl, and subsequently applied to a Superdex 200 increase 10/300 GL (Cytiva) gel filtration column. Fractions containing AF647-labeled C9 were collected and analyzed to have a AF647 dye to protein label (DOL) of 56%.

## Convertase labeling of bacteria in ΔC5 serum and MAC deposition

Bacteria were grown to mid-log phase as described above, washed and resuspended in RPMI/HSA. Bacteria with an $OD_{600\,nm}$ ~ 0.1 were incubated with 10% C5-depleted serum for 30 min, shaking at 37 °C. HADA-labeled bacteria were first grown for 15 min at 37 °C before adding C5-depleted serum. After two washing steps, convertase-labeled bacteria ($OD_{600\,nm}$ ~ 0.05) were incubated with a mix of 2.5% serum equivalent C5, C6, C7, C8(-AF488) and C9(-AF647) together with Sytox Blue (5 μM) for different amounts of time (indicated in the figures), shaking at 37 °C. At each indicated timepoint, a sample was taken out and immediately washed with RPMI/HSA. After the last timepoint was collected and washed, all samples were washed again and resuspended in RPMI/HSA to $OD_{600\,nm}$ ~ 0.5. These samples were directly applied on top of an agar pad for widefield imaging (described below). For flow

cytometry, samples were diluted to $OD_{600\,nm} \sim 0.005$ in RPMI/HSA with Sytox Blue (5 μM). When MAC deposition was induced on antibiotic-treated bacteria, 50 μg/ml chloramphenicol (Sigma-Aldrich), 50 μg/ml mecillinam (TargetMol) or 100 μg/ml cephalexin (Melford) was added during incubation with C5-depleted serum and during incubation with purified MAC components.

## Antibody-mediated MAC deposition in a purified classical pathway setup

Bacteria were grown to mid-log phase as described above, washed and resuspended in RPMI/HSA. Bacteria with an $OD_{600\,nm} \sim 0.1$ were incubated with 1 μg/ml anti-GlcNAc IgM and 2.5% serum equivalent C1 for 15 min, shaking at 4 °C. Next, a mix of C2, C3, C4 and C1-inhibitor was added to a final concentration of 2.5% serum equivalent and incubated for 15 min, shaking at 37 °C. Finally, a mix of C5, C6, C7, C8, C9-AF647 was added to a final concentration of 2.5% serum equivalent (C1 now being at 0.6% serum equivalent and C2, C3, C4 and C1-inh at 1.25% serum equivalent), together with Sytox Blue (5 μM). This mix was incubated for different amounts of time (indicated in the figures), shaking at 37 °C. At each indicated timepoint, a sample was taken out and immediately washed with RPMI/HSA. After the last timepoint was collected and washed, all samples were washed again and resuspended in RPMI/HSA to $OD_{600\,nm} \sim 0.5$. These samples were directly applied on top of an agar pad for widefield imaging (described below). For flow cytometry, samples were diluted to $OD_{600\,nm} \sim 0.005$ in RPMI/HSA with Sytox Blue (5 μM).

## IgM, C3b and C5 deposition

Bacteria were grown to mid-log phase as described above, washed and resuspended in RPMI/HSA. To measure anti-GlcNAc deposition, bacteria with an $OD_{600\,nm} \sim 0.1$ were incubated with 1 μg/ml anti-GlcNAc IgM for 30 min, shaking at 4 °C. For IgM detection, bacteria were washed and then incubated with 1.5 μg/ml Goat F(ab')2 anti-kappa-AF647 (Southern Biotech) for 30 min, shaking at 4 °C. To measure C3b deposition, bacteria with an $OD_{600\,nm} \sim 0.1$ were labeled with convertases by incubation with 10% C5-depleted serum for 30 min, shaking at 37 °C. Next, bacteria were washed and incubated with 3 μg/ml monoclonal mouse anti-C3b (bH6 (Garred et al, 1988)) that was randomly labeled with NHS-AF488 (Thermo Fisher Scientific) according to the manufacturer's protocol. For staining of C5, convertase-labeled bacteria that had been exposed to MAC components were washed and incubated with Goat Anti-Human C5 (Complement Technology). After washing, secondary staining was performed with AF647-conjugated Donkey Anti-Goat IgG (Jackson ImmunoResearch). Finally, bacteria were washed and resuspended in RPMI/HSA to $OD_{600\,nm} \sim 0.5$ for microscopy.

## Widefield microscopy

Agar pads were prepared from 1% low-melting (SeaPlaque) agarose (Lonza) in PBS or, for time-lapse imaging, in RPMI/HSA. A 22 mm × 22 mm × 0.96 mm siliconized glass cover slide (Hampton Research) was taped around the edged to create a chamber-forming coverslip. Heated agarose was pipetted on a microscope slide (Epredia) and covered with the chamber-forming coverslip to prepare the agar pad. If Sytox Blue was measured, Sytox Blue

(5 μM) was added to the heated agarose solution before putting it on the microscope slide. After solidification of the agarose solution the chamber-forming coverslip was removed. Bacteria at $OD_{600\,nm} \sim 0.5$ that had been exposed to the experimental conditions described above were applied on top of the agar pad and dried for ~10 min. at room temperature before they were covered with a 22 × 22 mm coverslip (Superior Marienfeld). Widefield microscopy was performed on a Leica SP5 microscope equipped with a HC PL APO100×/1.40 OIL PH3 objective. AF647, AF488, Sytox Blue and HADA were detected using a Y5, GFP, YFP or CFP filter cube. For time-lapse microscopy, the microscope environment was maintained at 37 °C and images were captured every 5–10 min for a period of 2.5 h. To calculate the number of divisions, we counted the number of bacteria derived from a single bacterium after 2.5 h, subtracting one.

## Bacterial viability

Convertase-labeled bacteria were incubated with purified C5-C9 and Sytox Blue for different amounts of time as described above. At each timepoint, samples were immediately washed and resuspended in RPMI/HSA. After the assay, samples were serially diluted in PBS and 20 μl droplets were plated in duplicate on LB agar plates. Colonies were counted after overnight incubation at 37 °C to calculate the number of colony forming units (CFUs) per ml in the original sample. Percentage survival was determined by comparison to the sample that had not been incubated with C5-C9 (0-min timepoint).

## Image analysis

Phase-contrast and fluorescence microscopy images of bacteria were analyzed using a custom Python (version: 3.11.9) script. The script was optimized for high-throughput analysis and supports multiple experimental conditions. It integrates deep learning models for instance segmentation (3-class U-Net (Caicedo et al, 2019; Ronneberger et al, 2015)) and object classification of bacterial growth stages (Ultralytics YOLO11 (Jocher et al, 2023)). In our tests, we found that the instance segmentation integrated into YOLO produces artifacts and poor segmentation results for small objects. To address this, a 3-class U-net was trained separately for instance segmentation. In general, using multiple smaller, specialized models provides better performance than a single large, generalized model (Kondratyuk et al, 2020).

## Deep learning models

Our 3-class U-Net was trained to predict the following classes: background, cell membrane (3px wide), cell body. The architecture is based on the residual convolutional neuronal network (adapted from pytorch-3dunet (Wolny et al, 2020)) combined with Spatial and Channel Attention (SCA, (Shan and Yan, 2021)) layers with 2,460,562 trainable parameters. We chose for our U-Net 5 layers with 16, 32, 64, 128, 256 channels, respectively. Each layer consists of SCA2D, GroupNorm, Convolution2D, ReLu activation with 2 × 2 max pooling for the down path and transposed convolution upscaling for the up path. A total of 80 images (73 for training and 7 for validation) were manually annotated using Napari (Sofroniew et al, 2024). Furthermore, full-size grayscale phase-contrast image

(1392 × 1040 pixels, 0.102 μm pixel size) and the annotated target images were cut into smaller 256 × 256 sub-images (n = 120) with 50% overlap. The model was trained with 27,594 training- and 2646 validation-image-sets (image augmentation to increase number of image-sets), Adam optimizer (weight decay = 1E-5, learning rate = 5E-4), Tversky and Focal loss using PyTorch (Ansel et al, 2024) and converged in 42 epochs (final validation metrics: accuracy: 0.994, IoU score: 0.982, Dice score: 0.991, F1: 0.991, recall: 0.991, training curves in Appendix Fig. S4B). Our YOLO model was trained to predict the following classes: dividing, rod-shaped, intermediate and defocused bacteria as well as microcolonies. We chose the yolo11m detection model from the Ultralytics YOLO11 implementation with 35,729,296 trainable parameters. A total of 99 images (76 for training and 23 for validation) with 3358 bounding boxes were manually annotated using a COCO annotator (Brooks, 2019) server instance (provided by the University of Applied Sciences Hagenberg), see annotation rules in Appendix Fig. S4A. The model was trained with default parameters (batch = 8, image size = 1024) for 400 epochs (converged after 239 epochs with final validation metrics of precision: 0.689, recall: 0.718, mAP50: 0.715, mAP50-95: 0.518, training curves in Appendix Fig. S4C and confusion matrix in Appendix Fig. S4D).

## Image processing pipeline and fluorescence intensity profile calculation

Each frame of a multidimensional image stack containing a phase-contrast and multiple fluorescence channels (e.g. C9-AF647, …) was successively analyzed. The phase-contrast channel for each frame (timepoint or spatial region) in an experiment was predicted by both deep learning models. For instance, segmentation using our 3-class U-Net model, first the full-size grayscale phase contrast image (1392 × 1040 pixels, 0.102 μm pixel size) was cut into smaller 256 × 256 sub-images (N = 120) with 50% overlap. The cut sub-images were then batched (typically batch size of 64 for 6GB of VRAM) and the 3-classes are predicted by the deep learning model. Thereby each pixel was categorized into a class depending on if the pixel contains a cell border, cell body or background. A full-size image (1392 × 1040 pixels) was then reconstructed by smooth blending overlapping predicted images (120 sub-images with 256 × 256 pixels). We used a second-order spline window function for blending with 50% overlap (Hauser et al, 2024). Next, cell contours were generated based on the cell body class using OpenCV's connected component function. Each contour was then rendered to an image again and dilated by exactly 3 pixels. The dilatation was necessary to compensate for the cell border class (which was always annotated to be 3 pixels wide). The reconstructed image (1392 × 1040 pixels) represents the instance segmentation mask for the currently processed single cell. The same phase-contrast image as for the U-Net was then used for object classification using our trained YOLO model. The model predicts multiple objects with a bounding box, classification (dividing, rod-shaped, intermediate, defocused bacteria, and microcolonies) and a classification confidence score. Our algorithm iterates over each cell mask from the segmentation step. Each instance segmented mask of a single cell was then correlated to a YOLO object by highest intersection over union (IoU) value and classification confidence score. Undesired bacterial growth stages (e.g. intermediate,

defocused bacteria, and microcolonies) and cell masks intersecting with the image borders were skipped and filtered out. The image was then cropped to the cell mask's bounding box with an additional small border (typically 20 pixels) around it. Based on this cell mask the long axis of the cell (longest central line or median axis) was then calculated. We offer several algorithms to find the long axis of a cell including skeletonization (Zhang and Suen, 1984), principal component analysis (Pedregosa et al, 2018), image moments (Hu, 1962) and elliptical fits. In our experience, the image moments gave the most consistent and reliable results of finding the long axis. The long axis was determined using the image moments algorithm by calculating the angle θ of the long axis from the central moments and cell center ($c_x$, $c_y$) using the spatial moments:

$$\theta = \frac{1}{2} tan^{-1} \frac{2m_{11}}{m_{20} - m_{02}}, c_x = \frac{m_{10}}{m_{00}}, c_y = \frac{m_{01}}{m_{00}}$$

Next, a line with the angle θ was projected out from the cell center until the border of the cell mask was reached. This long axis line was then fitted using a B-spline of degree 1 (linear fit) for further processing. N (typically 21) points were then equidistantly distributed along the long axis line from 5% to 95% of the total length (i.e. line length divided by 100, $p_1 = 5/100$, $p_2 = 9.5/100$, …, $p_{20} = 90.5/100$, $p_{21} = 95/100$). For each point, a perpendicular line intersecting with the cell mask was calculated and intensities values for all available fluorescence channels are sampled (typically using sub-pixel values and bilinear interpolation). A fluorescence intensity profile was then calculated by using the maximum pixel values along the sampled perpendicular line. Alternatively, the mean of the pixel values can be used. A comparison of maximum and mean fluorescence intensity profiles using simulated datasets can be found in Fig. EV3. These fluorescence intensity profiles containing N values as well as frame number, class, index, confidence, contour, length and area were saved in a Panda's data frame for each instance of a segmented cell. Based on the extracted values, the cells can be further processed to remove outliers or exclude cells with certain conditions (e.g., Sytox-negative). A length threshold can be calculated to filter out misclassified rod-shaped and dividing bacteria. We found that the Otsu threshold algorithm (Otsu, 1979) allows reliable separation of the two distributions based on the cell length histogram. Image thresholding was used to identify cells exhibiting certain conditions based on fluorescence intensities. A reliable threshold was determined by first applying a Gaussian blur (σ = 5 pixels) to the fluorescence channel image, followed by thresholding using the triangular algorithm (Zack et al, 1977). Cells whose cell mask overlapped with the thresholded image were regarded as positive. Cells lacking fluorescence signals above the calculated threshold within their masks were excluded from further analysis. A final box plot containing all the fluorescence intensity profiles by class (desired bacterial growth state, i.e., Rods, Dividing) was generated after all cells and frames were processed and filtered. Bacteria were oriented based on the C9 or HADA fluorescence channel: if the right half of a bacterium showed a higher intensity than the left half, the cell was flipped so that the higher signal was always positioned on the left (Appendix Fig. S11). For a direct comparison of different fluorescence channels, the profiles were normalized to 1.

## Average shapes

The average shape of classified (bacterial growth state: rod-shaped and dividing) and filtered bacteria was calculated based on the determined cell mask contours (findContours from OpenCV, which returns a list of vertex positions along the contour) from the image processing pipeline. The contours were then normalized between minus and plus one and rotated so that the long axis of the cell is parallel to the image x-axis. Normalization was facilitated by performing an elliptical fitting the contour positions (fitEllipse from OpenCV, which returns a rotated bounding box). In addition, the number of vertices along the contour of each cell was equalized by interpolation and/or simplification. Next, an average shape is optimized using PyTorch's ADAM optimizer, where no neuronal network was trained, but the average distance between all vertex positions was minimized. The contour with the lowest Hausdorff distance (van Kreveld et al, 2022) value was selected for an initial estimate of the average shape and iteratively refined. The mean Hausdorff distance was used as a loss function with a maximum of 1000 iterations, and the optimization was stopped early if the difference between current and previous loss was less than $10^{-8}$.

## Computation system architecture

Computations for bacteria analysis and development of algorithms were performed on a Dell Precision 3581 notebook (CPU: Intel Core i5-13600H with 12 cores at 2.8 GHz, RAM: 32 GB with a speed of 5200 MHz, GPU: NVIDIA RTX A1000 with 6GB VRAM, OS: Window 10 Education operating system 64 bit) and for deep learning on a Dell XPS 8950 workstation (CPU: Intel Core i7-12700k with 12 cores at 3.6 GHz, RAM: 64GB DDR5 4600MT/s, GPU: NVidia RTX 3060 with 12GB VRAM, OS: Windows 11 Home operating system 64 bit).

## Flow cytometry analysis

Flow cytometry was performed on a MACSQuant VYB flow cytometer (Miltenyi Biotec). ~10,000 bacterial events were recorded per sample. Flow cytometry data were analyzed in FlowJo (v.10). Bacteria were gated based on forward and side scatter (Appendix Fig. S1). Geometric Means of Fluorescence Intensity (GeoMFIs) were based on voltage height values. To plot the GeoMFI of C9-AF647 and Sytox Blue in the same graph, $\log_{10}$(GeoMFI) values were normalized by setting the minimum and maximum value at 0 and 100, respectively.

## Statistical analysis and graphs

Statistical analysis was performed in GraphPad Prism (v.9) as specified in the figure legends. Graphs were made using GraphPad Prism (v.9) and edited for appearance using Adobe Illustrator.

## Ethics statement

Human blood was isolated after informed consent was obtained from the subject in accordance with the Declaration of Helsinki. Approval was obtained from the medical ethics committee of the UMC Utrecht, the Netherlands.

## Data availability

The datasets and computer code produced in this study are available in the following databases: Modeling computer scripts: GitHub (https://github.com/CURTLab/BacFluoMap). Imaging data: BioImage Archive S-BIAD2294.

The source data of this paper are collected in the following database record: biostudies:S-SCDT-10_1038-S44319-025-00669-1.

## Peer review information

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

## Acknowledgements

The work was funded by the European Research Council (ERC) under the European Union's Horizon 2020 research and innovation program (grant agreement no. 101001937, ERC-ACCENT, to SHMR) and by the Austrian Forschungsförderungsgesellschaft (FFG, Nano-Carriers, project number: 898921). We would like to acknowledge Marek Basler, Johannes Schneider and Andrea Vettiger for the valuable technical input on live-cell imaging of bacteria and Jos van Strijp for proofreading of the manuscript. Furthermore, we would like to thank Floor Annabel Niessen for providing us with advice on statistical tests used.

## Author contributions

**Marije FL van 't Wout**: Conceptualization; Data curation; Formal analysis; Validation; Investigation; Visualization; Methodology; Writing—original draft; Writing—review and editing. **Fabian Hauser**: Software; Formal analysis; Validation; Investigation; Visualization; Methodology; Writing—original draft; Writing—review and editing. **Philippa I P Holzapfel**: Formal analysis; Validation; Investigation; Visualization; Writing—review and editing. **Bart W Bardoel**: Conceptualization; Resources; Investigation; Methodology; Writing—review and editing. **Carla J C de Haas**: Resources; Methodology; Writing—review and editing. **Jaroslaw Jacak**: Conceptualization; Software; Supervision; Funding acquisition; Investigation; Methodology; Writing—review and editing. **Suzan HM Rooijakkers**: Conceptualization; Supervision; Funding acquisition; Investigation; Methodology; Writing—review and editing. **Dani A C Heesterbeek**: Conceptualization; Formal analysis; Supervision; Validation; Investigation; Visualization; Methodology; Writing—original draft; Writing—review and editing.

Source data underlying figure panels in this paper may have individual authorship assigned. Where available, figure panel/source data authorship is listed in the following database record: biostudies:S-SCDT-10_1038-S44319-025-00669-1.

## Disclosure and competing interests statement

The authors declare no competing interests.

# Expanded View Figures

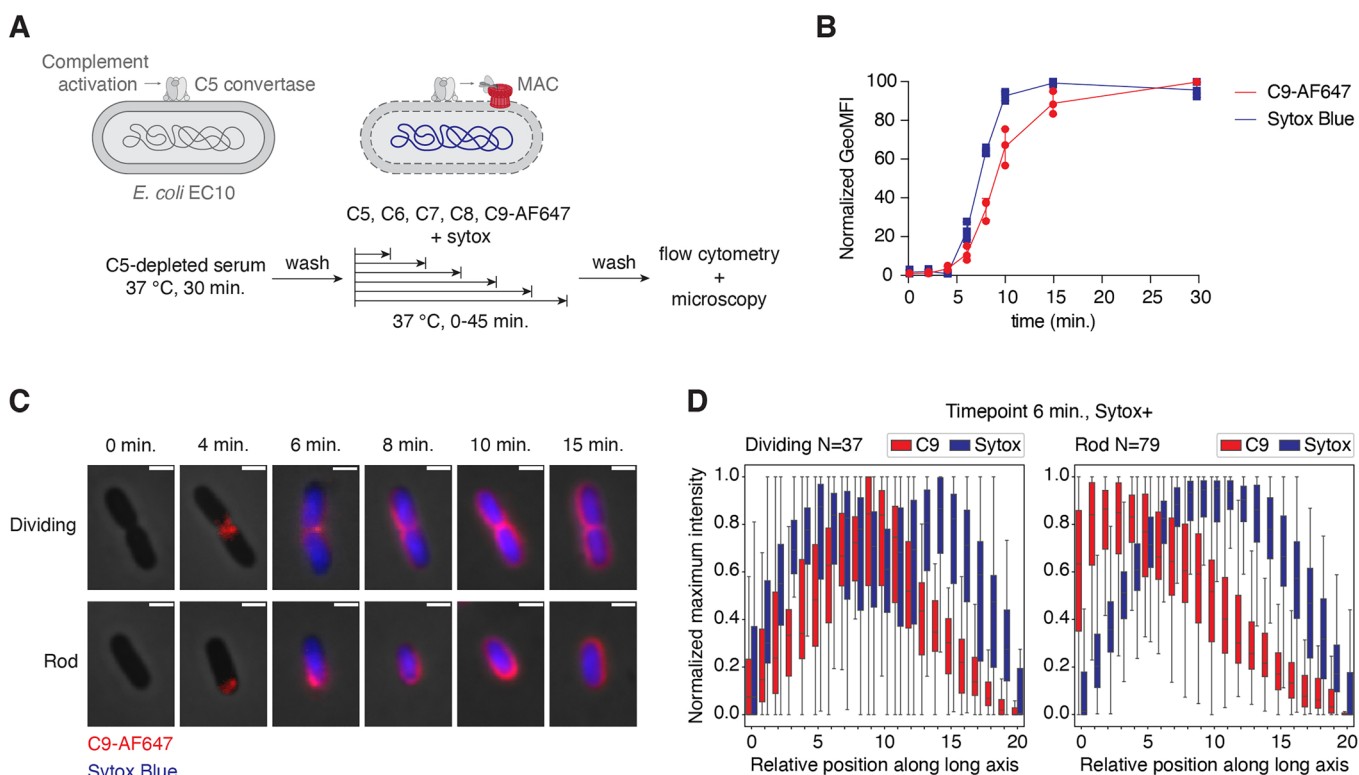

**Figure EV1. MAC localization on a clinical *E. coli* isolate.**

(A) Schematic overview of the experimental setup used to study MAC deposition and inner membrane in time on *E. coli* EC10. (B) C9-AF647 deposition and inner membrane damage (Sytox Blue) in time, measured by flow cytometry. (C) Phase-contrast and overlayed fluorescence images of selected bacteria at different timepoints. C9-AF647 is shown in red and Sytox in blue. (D) Normalized average C9 and Sytox distribution along the long axis of dividing and rod-shaped bacteria after 6 min incubation with C5-C9. Data information: In (B), GeoMFI values were normalized by calculating the percentage of the maximum value after log-transformation. Data represent individual values with mean ± SD of three biological replicates. In (C), individual bacteria have been chosen to reflect an average bacterium in the sample and are representative for three biological replicates. Scale bars: 2 μm. In (D), each box represents the interquartile range (IQR) of the data, with the center line indicating the median relative intensity. The whiskers extend to the most extreme values within 1.5× IQR. Data analysis was performed on all 14 images of one biological replicate, containing 37 dividing and 79 rod-shaped bacteria. Source data are available online for this figure.

**A**

anti-GlcNAc distribution

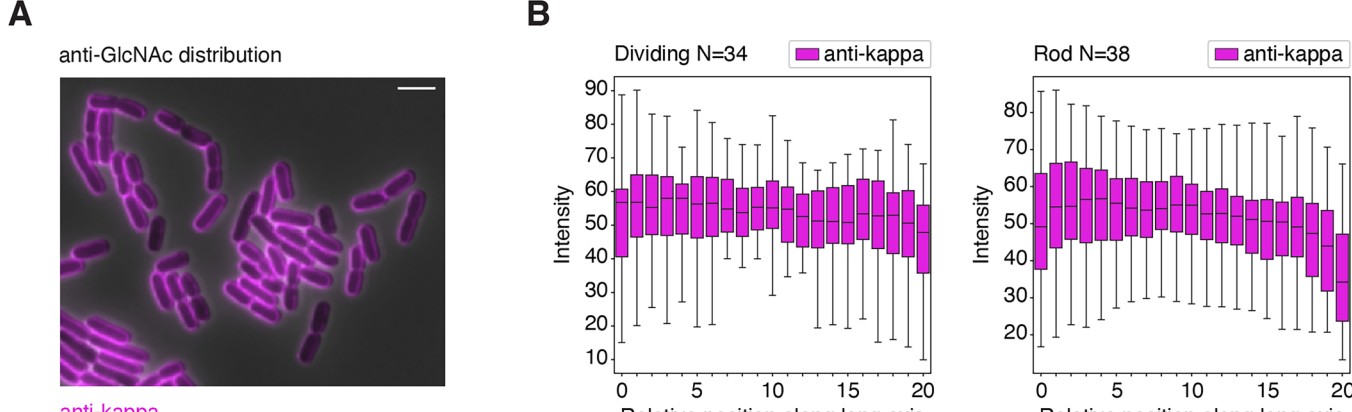

anti-kappa

**B**

Dividing N=34    ■ anti-kappa    Rod N=38    ■ anti-kappa

**Figure EV2.  Anti-GlcNAc IgM distribution on *E. coli*.**

(A) Phase-contrast and overlayed fluorescence images of bacteria that were incubated with 1 µg/ml anti-GlcNAc IgM followed by staining with a fluorescently labeled anti-kappa F(ab')2 fragment. (B) Average anti-GlcNAc IgM distribution along the long axis of dividing and rod-shaped bacteria. Data information: In (A), scale bar: 5 µm. In (B), each box represents the interquartile range (IQR) of the data, with the center line indicating the median relative intensity. The whiskers extend to the most extreme values within 1.5× IQR. Data analysis was performed on all ten images of one biological replicate, containing 34 dividing and 38 rod-shaped bacteria.

**A**

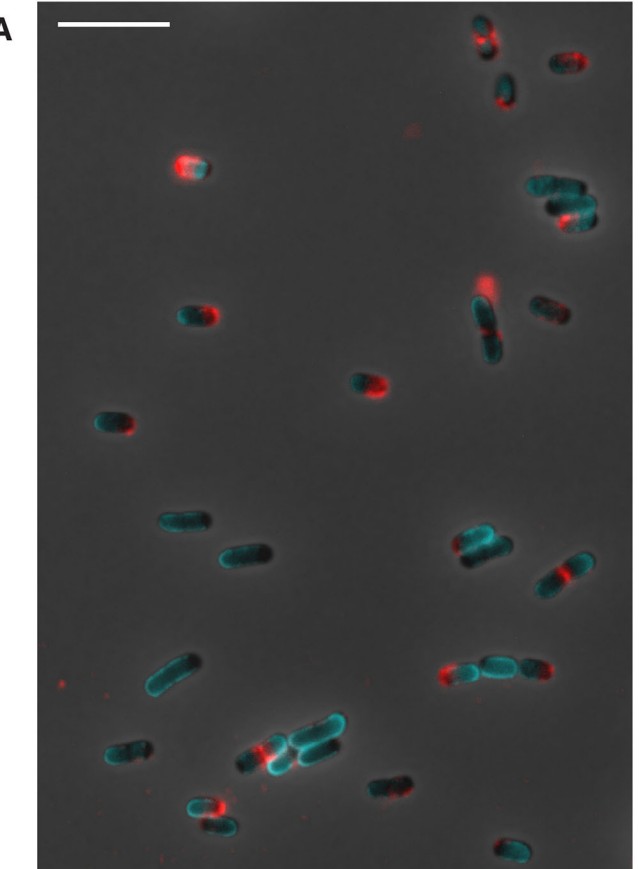

**B**

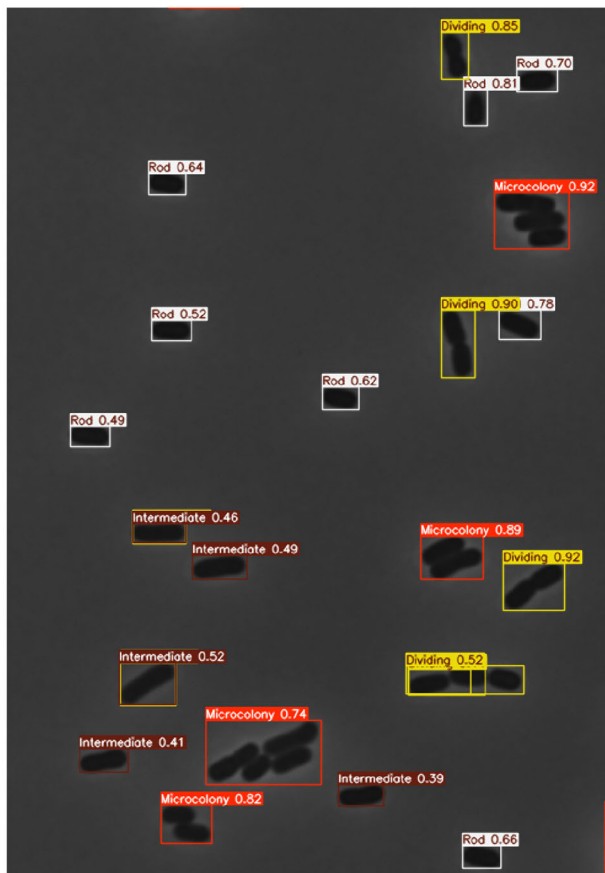

**C**

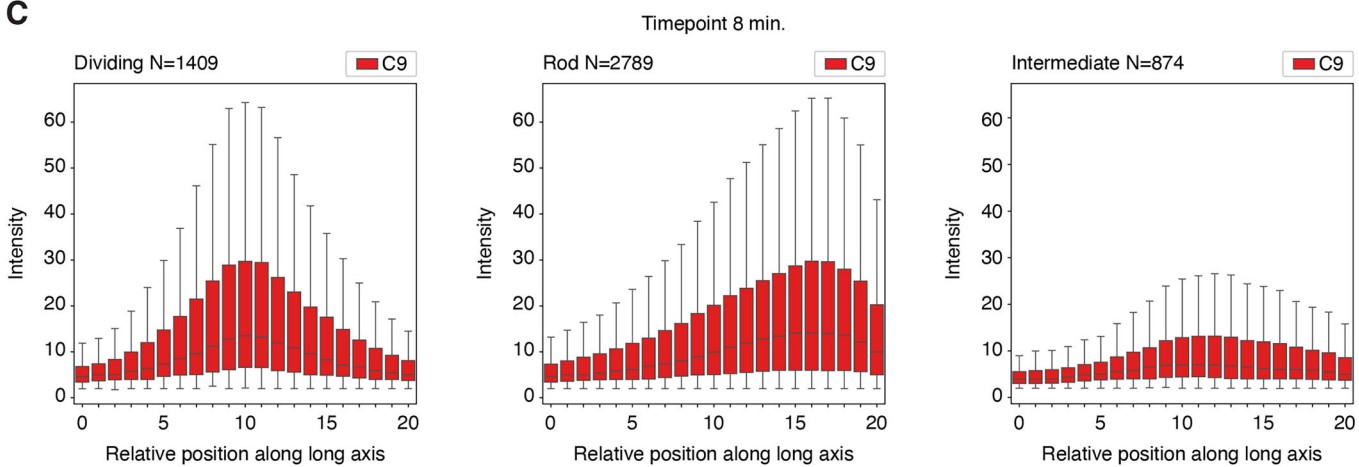

Timepoint 8 min.

**Figure EV3. Analysis of MAC localization on intermediate stage bacteria.**

(A) Phase-contrast and overlayed fluorescence images of bacteria that were treated as depicted in Fig. 3B. C9-AF647 is shown in red and HADA in cyan. (B) Growth stage classification of bacteria shown in (A). Dividing bacteria are in yellow boxes, rods in white boxes, intermediates in brown boxes and microcolonies in red boxes. (C) Average C9-AF647 distribution along the long axis of dividing, rod-shaped and intermediate stage bacteria. Data information: In (A), scale bar: 10 μm. In (C), each box represents the interquartile range (IQR) of the data, with the center line indicating the median relative intensity. The whiskers extend to the most extreme values within 1.5× IQR. Data analysis was performed on all 50 images of one biological replicate.

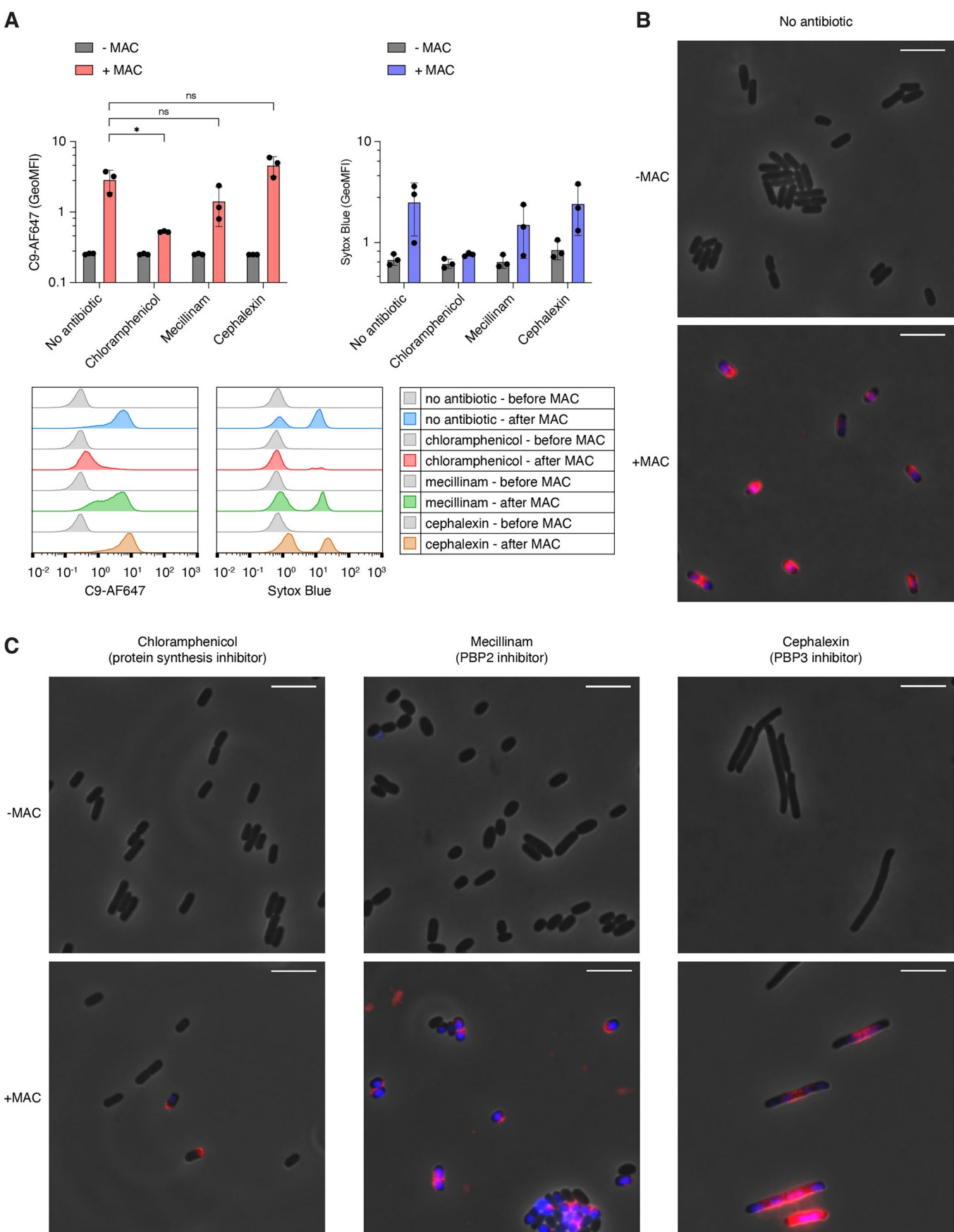

**Figure EV4.   MAC deposition on antibiotic-treated *E. coli*.**

(**A**) *E. coli* MG1655 was pre-incubated with C5-depleted serum, washed, and then incubated with purified MAC components for 8 min. Chloramphenicol, mecillinam or cephalexin were present during incubation with C5-depleted serum and during incubation with purified MAC components. Graphs show flow cytometry analysis of C9-AF647 and Sytox Blue fluorescence intensities before and after incubation with MAC components. (**B, C**) Widefield fluorescence images show bacteria before and after incubation with purified MAC components. C9-AF647 is shown in red and Sytox in blue. Data information: In (**A**), data represent individual values with mean ± SD of three biological replicates. Statistics was performed using a one-way ANOVA ($P = 0.0041$ for C9, $P = 0.1889$ for Sytox) followed by Dunnett's post-hoc test, comparing each group with the no antibiotic control. *$P < 0.05$. In (**B, C**), images are representative for three biological replicates. Scale bars: 10 μm. Source data are available online for this figure.

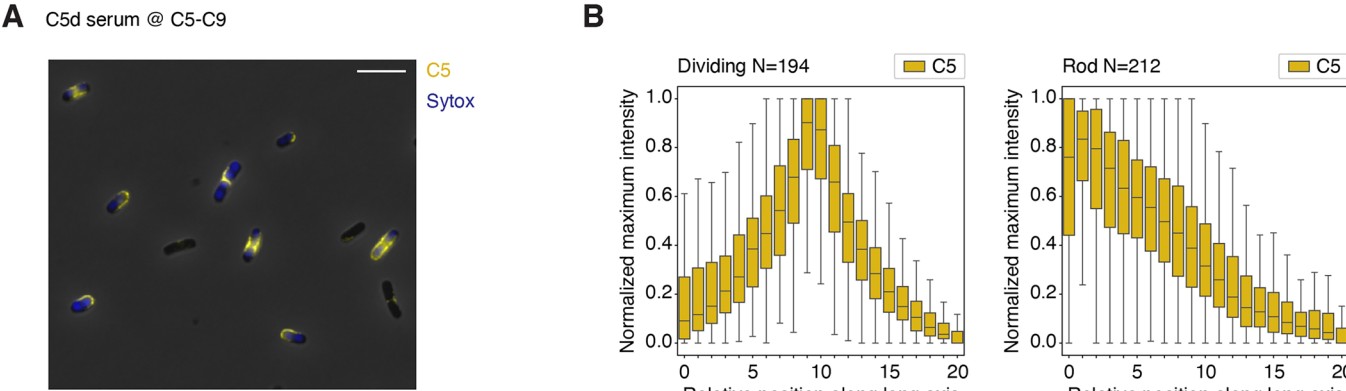

**A**  C5d serum @ C5-C9

**Figure EV5.  MAC component C5 also localizes at the new pole of *E. coli*.**

(**A**) Convertase-labeled *E. coli* bacteria were exposed to C5-C9 for 4 min followed by staining with anti-C5 and a fluorescently labeled secondary antibody. Localization of C5 (yellow) was determined by phase-contrast and widefield fluorescence microscopy. (**B**) Data analysis of the sample shown in (**A**). Graphs show the normalized average C5 distribution along the long axis of dividing and rod-shaped bacteria. Data information: In (**A**), scale bar: 10 µm. In (**B**), each box represents the interquartile range (IQR) of the data, with the center line indicating the median relative intensity. The whiskers extend to the most extreme values within 1.5× IQR. Data analysis was performed on all 30 images of one biological replicate, containing 194 dividing and 212 rod-shaped bacteria. The image and graphs are representative for three biological replicates.

