## [Peer Review File · EMBO Reports]

Bactericidal Membrane Attack Complex formation initiates at the new pole of *E. coli*

Marije van 't Wout, Fabian Hauser, Philippa Holzapfel, Bart Bardoel, Carla de Haas, Jaroslaw Jacak, Suzan Rooijakkers, and Dani Heesterbeek

Corresponding author(s): Dani Heesterbeek (D.A.C.Heesterbeek-2@umcutrecht.nl)

Review Timeline:

Submission Date:	29th Apr 25
Editorial Decision:	10th Jun 25
Revision Received:	9th Sep 25
Editorial Decision:	22nd Oct 25
Revision Received:	19th Nov 25
Accepted:	25th Nov 25

Editor: Achim Breiling

Transaction Report:

Dear Dr. Heesterbeek,

Thank you for the transfer of your manuscript to EMBO reports. I have now received reports from the three referees that were asked to evaluate your study, which can be found at the end of this email. As you will see, the referees think that these findings are of high interest. However, they have several comments, concerns, and suggestions, indicating that a major revision of the manuscript is necessary to allow publication of the study in EMBO reports. As the reports are below, and all the referee concerns need to be addressed, I will not detail them here.

Given the constructive referee comments, I would like to invite you to revise your manuscript with the understanding that the concerns of the referees must be addressed in the revised manuscript and/or in a detailed point-by-point response. Acceptance of your manuscript will depend on a positive outcome of a second round of review. It is EMBO reports policy to allow a single round of revision only and acceptance of the manuscript will therefore depend on the completeness of your responses included in the next, final version of the manuscript.

- 1) a .docx formatted version of the final manuscript text (including legends for main figures, EV figures and tables), but without the figures included. Figure legends should be compiled at the end of the manuscript text.
- 2) individual production quality figure files as .eps, .tif, .jpg (one file per figure), of main figures and EV figures. Please upload these as separate, individual files upon re-submission.

- 4) a complete author checklist, which you can download from our author guidelines

(<https://www.embopress.org/page/journal/14693178/authorguide>). Please insert page numbers in the checklist to indicate where the requested information can be found in the manuscript. The completed author checklist will also be part of the RPF.

- 5) that primary datasets produced in this study (e.g. RNA-seq, ChIP-seq, structural and array data) are deposited in an appropriate public database. If no primary datasets have been deposited, please also state this in a dedicated section (e.g. 'No

primary datasets have been generated and deposited'), see below.

The accession numbers and database should be listed in a formal "Data Availability" section that follows the model below. This is now mandatory (like the COI statement). Please note that the Data Availability Section is restricted to new primary data that are part of this study. This section is mandatory. As indicated above, if no primary datasets have been deposited, please state this in this section

Data availability

6) We now request the publication of original source data with the aim of making primary data more accessible and transparent to the reader. You will receive a separate email with instructions for providing source data with your revised manuscript, including information how to upload and organize the files.

8) Regarding data quantification and statistics, please make sure that the number "n" for how many independent experiments were performed, their nature (biological versus technical replicates), the bars and error bars (e.g. SEM, SD) and the test used to calculate p-values is indicated in the respective figure legends (also for EV and Appendix figures). Please also check that all the p-values are explained in the legend, and that these fit to those shown in the figure. Please provide statistical testing where applicable. Please avoid the phrase 'independent experiment', but clearly state if these were biological or technical replicates. Please also indicate (e.g. with n.s.) if testing was performed, but the differences are not significant. In case n=2, please show the data as separate datapoints without error bars and statistics. See also: <http://www.embopress.org/page/journal/14693178/authorguide#statisticalanalysis>

9) Please add scale bars of similar style and thickness to all microscopic images, using clearly visible black or white bars (depending on the background). Please place these in the lower right corner of the images themselves. Please do not write on or near the bars in the image but define the size in the respective figure legend.

10) Please also note our reference format:
<http://www.embopress.org/page/journal/14693178/authorguide#referencesformat>

12) We now use CRediT to specify the contributions of each author in the journal submission system. CRediT replaces the author contribution section. Please use the free text box to provide more detailed descriptions and do NOT provide your final manuscript text file with an author contributions section. See also our guide to authors: <https://www.embopress.org/page/journal/14693178/authorguide#authorshipguidelines>

13) All Materials and Methods need to be described in the main text using our 'Structured Methods' format, which is required for all research articles. According to this format, the Methods section should include a Reagents and Tools Table (listing key

reagents, experimental models, software, and relevant equipment and including their sources and relevant identifiers), uploaded as separate file, and a Methods section in which we encourage the authors to describe their methods using a step-by-step protocol format with bullet points, to facilitate the adoption of the methodologies across labs. More information on how to adhere to this format as well as downloadable templates (.doc) for the Reagents and Tools Table can be found in our author guidelines (section 'Structured Methods'):

14) Please reduce the keywords to five and order the sections like this, using these names:

Title page - Abstract - Keywords - Introduction - Results - Discussion - Methods - Data availability section - Acknowledgements (please include here also the funding information) - Disclosure and Competing Interests Statement - References - Figure legends - Expanded View Figure legends

15) Please make sure that all the funding information is also entered into the online submission system and that it is complete and similar to the one in the acknowledgement section of the manuscript text file.

I look forward to seeing a revised form of your manuscript when it is ready.

Yours sincerely,

Referee #1:

The manuscript by M van't Wout et al presents an exciting study investigating the drivers behind MAC-mediated bacterial lysis during complement activation. They use live cell imaging and flow cytometry to understand where MAC deposits on bacterial cell surface and how this influences MAC activity. It builds on a growing excitement in immunology and provides the next key discovery in the field. The manuscript is well written, figures clearly communicate the findings, and the conclusions are broadly justified by the data presented. I have a few questions and suggestions on how to improve clarity but overall am positive on the manuscript and am of the opinion it should be published subject to minor revision.

The first figure shows that MAC is preferentially formed at the septum of dividing cells and at the new pole of the growing bacterium. They show that MAC deposition on lab-adapted *E.coli* is similar to that found on clinical isolates.

Key to their study is their development of computational tools and image processing pipeline to analyse fluorescence data and identify cells in specific states of cell division based on cell shape. 1) what advantage does their new software tool offer over what is commercially available in cell profiler? 2) I don't think I saw a link or call out for the software to be made publicly available? Please provide a github link for the python script etc upon publication in line with open source/data availability norms. 3) could the authors say anything about data exclusions for the collected images?

They next use a specific antibody to activate the classical pathway on the bacterium and see the same distribution pattern.

Figure 3 links the deposition of MAC to PG synthesis. They use a fluorescent version of a d-alanine which incorporates into the PG as a timestamp that distinguishes new vs old poles. Their data show that MAC is found preferentially in the pole containing the newly synthesized PG. One question I had was why is there an uneven intensity of HADA between the two "old" poles of the dividing cell in C? Based on the schematic in A I would have expected there to be blue on both ends?

The next part in Figure 4 I found a little confusing. It is about MAC deposition at the new poles impairing cell division/growth.

They use sytox blue as a measure of inner membrane damage which is typically a cell death stain. Is the argument they are making that inner membrane permeabilization doesn't necessarily mean cell death? How much is too much then? How would you determine/define cell death then?

Figure 5 shows that although MAC deposition initially happens at the septum and new pole, C3b deposition is uniform. The authors use C3b deposition as a proxy for C3 convertase, but what about the C4b derived convertase?

The Discussion section is well argued. Perhaps one thing that could add value is a discussion about differences in lipid compositions at these sites of MAC insertion.

The clarity of the methods could be improved to ensure reproducibility of the results. There are a few comments for each section/experiment type:

Convertase labelling of bacteria in $\Delta C5$ serum and MAC deposition

Clarification on the incubation time after the addition of terminal pathway protein. Are the incubations followed by a wash step? Or only at the end to remove excess fluorophore?

What OD600 reading of the bacteria for before the flow and microscopy experiments?

Why use RPMI with HSA when resuspending the cells (as opposed to PBS)?

Flow cytometry analysis

What is the total number of events recorded?

The gating strategy should be included as supporting data.

Protein expression and fluorescent labelling

For the Fluorescent proteins degree of labelling (DOL), the authors use a % for C9 and a ratio for C8 - these should be consistent.

How is His-TEV-sortase produced, or is it commercial?

The authors describe the expression of C9 and C8 as in reference 29, but this reference is for expression of C5. The methods for this paper should be self-contained.

Antibody mediated MAC

What were the incubation times for the addition of C2, C3, C4 and C1-inhibitor, and were they added together or sequentially?

Are expi293 cells used to express C2 or is C2 commercially sourced? In the reagents list, they have both? Which is one used vs the other?

When forming MAC, were the terminal pathway proteins added together, sequentially and how long were they incubated for?

Widefield microscopy

Which low melting agarose they use - include in reagent list cat number, company etc?

What glass slides and chamber forming cover slip were used - please include in the reagent list. Is there a reference to the agar pad method? This isn't clear how they were made.

How were the bacteria dried on top of the pad; temperature, duration?

How were the serum and complement components added for imaging, this is not stated in the methods?

Referee #2:

This elegant study describes how localized assembly of the membrane attack complex (MAC) disrupts the cell envelope and kills Gram-negative bacteria. Using time-lapse microscopy of *E. coli* with fluorescently labeled complement proteins and peptidoglycan, they provide convincing evidence that at early times and concentrations of complement, MAC is found preferentially at the cell division septum. After division, MAC remains at the new pole of *E. coli*. MAC localization is distinct from C3b, which is found across the bacterial cell. An additional strength of this work is the development and use of an algorithm to quantify the distribution of fluorescence of complement proteins or peptidoglycan along the length of *E. coli*. This study is an important addition to the understanding of how MAC pore formation leads to bacterial death.

Major comments:

1) The major unanswered question in this work is how MAC becomes localized to the division septum. Use of a fluorescently labeled anti-C5(b) antibody or fluorescently labeled C5 (if fluorophore is intact after C5 convertase cleavage) in C5-depleted serum would address whether this happens early in MAC assembly. It is interesting that although C8 mostly colocalizes with C9, it is less targeted to the pole/division septum than C9 (Figure 5). Could C9, with its membrane insertion domain, enable the targeting of MAC? This point could be mentioned in the Discussion (does not require an experiment).

2) Does MAC deposition require full cell division and separation, or are the cell division-associated changes in PG or OM composition sufficient to drive MAC localization? Evaluating MAC in *E. coli* treated with PG inhibitors (like mecillinam, to cause filamentation) compared with inhibitors of bacterial growth patterning (like A22) would address this key question.

Minor comments for the authors' consideration:

1) Please describe in the Results section how the algorithm developed here accounts for out-of-focus fluorescence on the bacterium.

2) In all figure legends, in addition to the number of images per experiment, please add the total number of bacteria that were analyzed. These numbers are a strength of the study.

3) In Figure 2, please explain why C1-inh was added for in vitro classical complement deposition.

4) For Figure 3, please add a supplemental figure showing the change in location of HADA fluorescence in growing *E. coli* with pulse-chase.

5) The content of Movies EV1-EV4 is not defined; please add. If Figure EV6 is an image from one of the movies, indicate that in the figure legend.

6) Figure 4C: please make symbols bigger, or remake the graph to better emphasize the time-dependent changes for Sytox positivity and C9 position.

Referee #3:

In their manuscript, van't Wout et al. study the bactericidal action of the MAC complex using a reconstituted system consisting of purified and fluorescently labeled MAC components. This setup allows the authors to track the fate of live bacterial cells exposed to MAC complex formation in a reproducible and highly controllable manner. Using this powerful methodology, the authors find that MAC complexes localize to the septum or new pole of newborn *E. coli* cells at short exposure times, accompanied by rapid disruption of the inner bacterial membrane. To quantify this effect, they developed a sophisticated, deep learning assisted image analysis pipeline that allows for classification into dividing and non-dividing ("rod") cells and quantify the distribution of the fluorescence signal along the bacterial length axis. Using this pipeline, the authors study the dynamics of the MAC complex formation following different pathways, concluding that the localized formation at early time points might be targeting weak spots (areas of cell wall synthesis) of the cell. While the presented work is solid and observations are well described, it lacks mechanistical insight. There are several points that should be addressed to strengthen and verify the hypotheses presented in the manuscript. I hope that the comments are helpful to the authors.

Major points:

1) Underlying cell wall synthesis model

The cell wall synthesis model employed by the authors and is based on polar PG synthesis. While recent studies have shown that polar PG is predominant during septation, PG insertion mainly occurs at the cell cylinder during cell elongation. This mode of cell wall synthesis is neither discussed by the authors nor taken into account during image analysis (neglecting of "intermediate" cells, see next point). Cell elongation is more prominent during fast growth, while septation is predominant under slow growth conditions. The authors should provide information on the doubling time (either mass doubling time in OD measurements or single-cell doubling times from time-lapse images) to determine the predominant PG synthesis pathway.

2) Classification of growth stages

The authors classify individual bacteria into the categories "rod", "dividing", "intermediate", "defocused" and "microcolonies". The category "rod" should be changed to "non-dividing", as all cells are actually rod-shaped. Moreover, it is not clear to me, which criteria were applied to discriminate "rod" from "intermediate" cells. The authors state that only bacteria formed after cell division were classified as "rod-shaped", which limits the investigation of MAC formation to dividing and newborn cells. What happens with cells that "repaired" the weak spots after cell division, i.e. cells that were in the elongation phase when MAC components were added? Do these cells show less fluorescence? Quantifying fluorescence intensities for "intermediate" cells might provide important information with respect to the "weak spot" hypothesis. If only small cells were classified as "rod", the polar MAC distribution is not surprising, as it is a result of the completion of cell division events after MAC addition. In any case, the criteria

for classification need to be specified and added to the manuscript.

3) Quantification of fluorescence images

The overall pipeline presented for single-cell analysis is innovative well designed. However, more information can be extracted from the data presented in the manuscript. The authors refer to the amount of MAC complexes in multiple sections of the manuscript without providing quantitative measures. As instance masks are available from the U-Net segmentation, it would be rather simple to quantify the integrated intensities per cell for the labeled subunits, providing a measure for the amount of pores. Such an analysis should be performed throughout the manuscript, as it allows the quantification of #pores vs time, #pores vs. cell size, #pores per growth stage, etc. I think that the recorded data is very interesting and that additional analyses could provide evidence for some of the postulated hypotheses. For example, on page 9 the authors state that the number of cell division correlated to the amount of MAC pores. However, there was no quantitative evidence provided other than a binary assessment (signal/no signal; growth/no growth), although the authors refer to Fig. EV6 for such an assessment. In the discussion (top lines on page 14), it is stated that bacteria with limited MAC deposition were able to recover. Here, proper quantification could provide a relative threshold of MAC pores that has to be reached for efficient killing.

4) Novelty of the mechanism

In the discussion, the authors state that polar MAC deposition was already observed in gram-positive bacteria in 2013. This raises the question about novelty of the observed phenotype. Can the authors highlight the advantages of their approach and discuss their results in the context of the mentioned study?

5) Mechanism of polar/septal MAC pore formation

The current work provides a solid description of MAC formation dynamics during early time points and its correlation with inner membrane damage. However, mechanistic insights into this process are hardly provided. For example, the authors state in the abstract that the "critical event causing MAC-mediated inner membrane damage remains elusive". This question is not addressed by the current work. Another crucial question is why MAC pores form at the new cell pole. Can the authors speculate on this? Could lipid composition play a role? The authors suggest that the new pole represents a "weak spot", but this hypothesis is only supported by similar patterns compared to nascent OMPs or septal PG synthesis. Regarding the observed localized MAC deposition, the authors pose several questions that could be tested directly. One question is whether active division is required for MAC-dependent killing. This question might be investigated using antibiotics that inhibit cell division (e.g. cephalixin). Another question is whether the increased insertion of OMPs at the division site enhances MAC deposition. This could be tested by inhibiting translation (e.g. using chloramphenicol) before and while adding MAC components. Another factor that could affect localization is cell curvature. Some proteins are known to be selective for positive or negative curvature (e.g. MreB, Min proteins) as curvature affects lipid bilayer packing. Could this also be a reason for the polar MAC deposition? The effect of curvature could be tested using antibiotics that affect cell elongation, such as A22 (MreB inhibitor) or Mecillinam (PBP2 inhibitor). It would be interesting to see if a morphological change affects MAC deposition.

6) Are non-polar MAC pores required for killing?

The CFU tests performed in this study showed that short incubation times that coincide with polar MAC deposition result in partial cell survival, while longer exposure with full deposition leads to complete killing. Isn't this an indication that non-polar deposition is at least partially contributing to killing?

7) Killing in complement-activated cells

The authors found that MAC-mediated killing takes place faster in complement-activated cells (Figure 2). While C9 is deposited earlier compared to non-IgM treated cells, inner membrane disruption dynamics are identical. This discrepancy is also seen in the representative cells (compare Fig. 1D vs. Fig. 2C). Do the authors have any explanation for this?

8) Statistics

While CFU tests and flow cytometry experiments were conducted in biological replicates, while some imaging experiments were based on technical replicates (several images of one individual experiment). Can the authors comment on reproducibility of these experiments? Why didn't the authors perform these experiments in replicates?

9) Selection of representative bacteria

The manuscript only provides a few exemplary cells. To get a better impression of cellular heterogeneity, galleries of several cells for each condition/growth phase could be provided in the supplementary information. At best, these cells originate from different replicates to support the statement that the selected cells in the main text are "representative for the average cell".

10) Validation/quality control of the trained deep learning models

The authors trained several deep learning models that were used for the image analysis. Typically, such models are tested in quality control experiments, in which their performance is quantified using suitable metrics. In case of segmentation and classification, this can be precision and recall or advanced metrics as dice index, F1-score or the panoptic quality. The authors should quantify model performance on unseen ground truth data and provide add the results to the manuscript.

11) Data availability

The deep learning networks, trained models and the image analysis pipelines are very interesting. It would be great if the

authors provide access to these resources by depositing them on suitable repositories (GitHub, Zenodo, etc.)

12) Comprehensive model

The experiments in this study give rise to MAC deposition dynamics and study the role of different pathways of the complement system. The findings/observations of this work could be well summarized in a comprehensive model. Such a model would assist the reader during the discussion and summarize the main points of this manuscript.

Other points:

- Page and line numbers should be added to simplify the review process
- Introduction: What does "single membrane particles" mean?
- The time points (0, 2, 4 ... X min) refer to time when cells were sampled and proceeded for washing. How long were the centrifugation steps during washing? Don't they add to the treatment duration? Experimental conditions for washing should be added to the manuscript.
- How was the number of cell divisions in Fig. 4 determined?
- The authors should provide the number of cells underlying the different experiments/conditions/time points in the figure caption.
- Time series shown in the supplementary videos could be stabilized (e.g. using the StackReg plugin in Fiji)
- Scale bars and time stamps should be added to the supplementary videos.
- Was Sytox Blue also added for the control experiment in Movie EV?
- The original U-Net paper by Ronnefeld et al. should be cited.
- The data analysis is advanced and well-constructed. Thus it should be acknowledged more throughout the manuscript.

Revised manuscript of "Bactericidal Membrane Attack Complex formation initiates at the new pole of *E. coli*" – reference EMBOR-2025-61837-T

Utrecht, September 9th, 2025

Dear Achim Breiling, dear editors of EMBO Reports,

Hereby we send you our revised manuscript entitled "**Bactericidal Membrane Attack Complex formation initiates at the new pole of *E. coli***". We thank you for your interest in our study and we are pleased to read that also the reviewers are excited about the results described in the manuscript. We also appreciate the critical evaluation of our study by the referees and have now extensively revised the manuscript based on their valuable suggestions. The revised manuscript contains 12 new figures (3 figures with new experimental data, 2 figures with additional data analysis, 6 technical figures and a comprehensive model). These include (but are not limited to) more extensive analysis of the mechanism behind the preferred localization of MAC pores on the new pole of *E. coli* via:

- Localization analysis of MAC deposition on bacteria in the intermediate growth stage category (Fig. EV3).
- Analysis of MAC localization on bacteria that were treated with antibiotics that interfere with cell division processes (Fig. EV4).
- Analysis of the localization of the first MAC component C5 (Fig. EV5).

Furthermore, we have included textual revisions in the text. In the point-to-point reply below, we answer the points that were addressed by the reviewers and indicate how we have included these comments in the manuscript. The main changes to the original manuscript are highlighted in yellow.

We trust that our revised version of the manuscript is now suitable for publication in EMBO Reports.

Sincerely, on behalf of the authors,
Dr. Dani Heesterbeek
University Medical Center Utrecht
Medical Microbiology, G04.614
Heidelberglaan 100, 3584 CX
Utrecht, The Netherlands
d.a.c.heesterbeek-2@umcutrecht.nl

POINT-TO-POINT REPLY

Referee #1:

The manuscript by M van't Wout et al presents an exciting study investigating the drivers behind MAC-mediated bacterial lysis during complement activation. They use live cell imaging and flow cytometry to understand where MAC deposits on bacterial cell surface and how this influences MAC activity. It builds on a growing excitement in immunology and provides the next key discovery in the field. The manuscript is well written, figures clearly communicate the findings, and the conclusions are broadly justified by the data presented. I have a few questions and suggestions on how to improve clarity but overall am positive on the manuscript and am of the opinion it should be published subject to minor revision.

The first figure shows that MAC is preferentially formed at the septum of dividing cells and at the new pole of the growing bacterium. They show that MAC deposition on lab-adapted *E. coli* is similar to that found on clinical isolates.

Key to their study is their development of computational tools and image processing pipeline to analyse fluorescence data and identify cells in specific states of cell division based on cell shape.

Point 1: “What advantage does their new software tool offer over what is commercially available in cell profiler?”

Answer 1:

Cell profiler is an open-source tool for automatic image processing and cell profiling including instance segmentation. We decided to develop our own pipeline since we could not find any tool that satisfied our needs: combination of robust instance segmentation of the bacterial growth stage using classification with fluorescence mapping along the long axis of bacteria. We hope that our annotated training set can be used to further improve future segmentation/classification deep learning models.

Point 2: “I don't think I saw a link or call out for the software to be made publicly available? Please provide a github link for the python script etc upon publication in line with open source/data availability norms.”

Answer 2:

The github repository is ready, included in the manuscript and will be opened upon publication including the used models and raw data to replicate the analysis.

Point 3: “Could the authors say anything about data exclusions for the collected images?”

Answer 3: For each subfigure showing fluorescence intensity distributions, we analyzed all images from one independent experiment. Although no entire images were excluded, clustered or out-of-focus bacteria identified by our deep learning algorithm were excluded from analysis. As we did not clearly state this in the text, this information is now added to the results section.

They next use a specific antibody to activate the classical pathway on the bacterium and see the same distribution pattern.

Figure 3 links the deposition of MAC to PG synthesis. They use a fluorescent version of a d-alanine which incorporates into the PG as a timestamp that distinguishes new vs old poles. Their data show that MAC is found preferentially in the pole containing the newly synthesized PG.

Point 4: “One question I had was why is there an uneven intensity of HADA between the two "old" poles of the dividing cell in C? Based on the schematic in A I would have expected there to be blue on both ends?”

Answer 4: The observation that HADA is primarily located at one of the two old poles is indeed surprising, as we expected symmetric growth of the two future daughter cells. Whereas more in-depth analysis of why this occurs is an interesting follow-up question, it is in our opinion beyond the scope of this manuscript. We did however include a supplementary figure in which we follow HADA localization over time (Appendix Figure S7). This shows that directly after incubation of bacteria with C5-depleted serum, HADA is distributed relatively homogeneously over the cell surface, and moves towards the old poles upon bacterial growth. After approximately 45 minutes, some bacteria may already have undergone a division cycle, which could account for the observation that HADA localizes predominantly to one side of the cell.

Point 5: “The next part in Figure 4 I found a little confusing. It is about MAC deposition at the new poles impairing cell division/growth. They use sytox blue as a measure of inner membrane damage which is typically a cell death stain. Is the argument they are making that inner membrane permeabilization doesn't necessarily mean cell death? How much is too much then? How would you determine/define cell death then?”

Answer 5: Although inner membrane damage is a good indicator of bacterial cell death when measuring the average of an entire bacterial population, we here wanted to focus on the heterogeneity in MAC deposition on individual bacteria and how this correlates with bacterial survival. We show that Sytox-positive bacteria can sometimes survive, but these bacteria likely only have low levels of inner membrane damage, as shown by a low Sytox intensity. We also showed previously that Sytox intensity correlates with bacterial survival (Heesterbeek et al, the EMBO Journal (2019)), with a high sytox signal indicating cell death, which we here confirm on individual bacteria.

Point 6: “Figure 5 shows that although MAC deposition initially happens at the septum and new pole, C3b deposition is uniform. The authors use C3b deposition as a proxy for C3 convertase, but what about the C4b derived convertase?”

Answer 6: We would like to clarify that we measured C3b deposition as a readout for complement activation and not for the localization of active C3 convertases. We have now also explained this more clearly in the manuscript. The finding that C3b molecules have been deposited all around the bacterial surface indicates that MAC localization is not determined by enhanced complement activation at certain locations. It would however, like the reviewer suggests, be interesting to measure active convertases (also those containing C4b molecules) in the future.

Point 6: “The Discussion section is well argued. Perhaps one thing that could add value is a discussion about differences in lipid compositions at these sites of MAC insertion.”

Answer 6: We now added some sentences to the discussion about the lipid composition of the bacterial outer membrane and how this could influence MAC localization.

Point 7: “The clarity of the methods could be improved to ensure reproducibility of the results.

There are a few comments for each section/experiment type:

Convertase labelling of bacteria in Δ C5 serum and MAC deposition

- Clarification on the incubation time after the addition of terminal pathway protein. Are the incubations followed by a wash step? Or only at the end to remove excess fluorophore?
All washing steps are now clarified in the methods section, including washing conditions. Bacteria were washed after incubation in C5-depleted serum and after incubation with MAC components.
- What OD600 reading of the bacteria for before the flow and microscopy experiments?
Before each experiment, bacteria were grown until mid-log phase (OD600nm \sim 0.4-0.6). We used bacteria at OD \sim 0.5 for microscopy and at OD \sim 0.005 for flow cytometry
- Why use RPMI with HSA when resuspending the cells (as opposed to PBS)?
Although we expect that resuspension in PBS would have given the same results, we decided to not change the medium in the last step to ensure live/dead discrimination with Sytox was not influenced by exchange of the medium.

Flow cytometry analysis

- What is the total number of events recorded?
 \sim 10,000 events
- The gating strategy should be included as supporting data.
This is now added a supplementary figure (Appendix Fig. S1).

Protein expression and fluorescent labelling

- For the Fluorescent proteins degree of labelling (DOL), the authors use a % for C9 and a ratio for C8 - these should be consistent.
Both DOLs are now expressed as a %.
- How is His-TEV-sortase produced, or is it commercial?
His-TEV-sortase was produced, which is now explained in the methods section.
- The authors describe the expression of C9 and C8 as in reference 29, but this reference is for expression of C5. The methods for this paper should be self-contained.
Expression of all purified complement components is now explained in detail in the methods section.

Antibody mediated MAC

- What were the incubation times for the addition of C2, C3, C4 and C1-inhibitor, and were they added together or sequentially?
Bacteria with bound antibodies were incubated with a mix of C2, C3, C4 and C1-inh for 15 minutes. Next, a mix of C5-C9 was added. No washing steps were performed between incubations with antibody, C2-C4 and C5-C9. This is now clarified in the methods section.
- Are expi293 cells used to express C2 or is C2 commercially sourced? In the reagents list, they have both? Which is one used vs the other?
C2 is always expressed in Expi cells.
- When forming MAC, where the terminal pathway proteins added together, sequentially and how long were they incubated for?
Terminal pathway proteins (C5-C9) were added together as a mix for different amounts of time, as specified in the figure legends.

Widefield microscopy

- Which low melting agarose they use - include in reagent list cat number, company etc?
We used SeaPlaque agarose from Lonza (50101).
- What glass slides and chamber forming cover slip were used - please include in the reagent list.

22mm x 22mm x 0.96mm siliconized glass cover slides (Hampton Research, HR3-233) were used to prepare chamber-forming coverslips. Glass slides are from EpreDia (15545650).

- Is there a reference to the agar pad method? This isn't clear how they were made.
We do not have a reference for the agar pad method, but we now provide a detailed description of the agar pad preparation in the methods section.
- How were the bacteria dried on top of the pad; temperature, duration?
Bacteria were dried on top of the pad for ~10 minutes at room temperature.
- How were the serum and complement components added for imaging, this is not stated in the methods?
For all experiments, bacteria were first incubated with serum and/or complement proteins in an Eppendorf tube. Only after preparation of the samples, bacteria were put on an agar pad and imaged. The same samples were used for flow cytometry and microscopy. We hope we have clarified this in the materials and method section.

Answer 7: All points mentioned above are now addressed in the manuscript.

Referee #2:

This elegant study describes how localized assembly of the membrane attack complex (MAC) disrupts the cell envelope and kills Gram-negative bacteria. Using time-lapse microscopy of *E. coli* with fluorescently labeled complement proteins and peptidoglycan, they provide convincing evidence that at early times and concentrations of complement, MAC is found preferentially at the cell division septum. After division, MAC remains at the new pole of *E. coli*. MAC localization is distinct from C3b, which is found across the bacterial cell. An additional strength of this work is the development and use of an algorithm to quantify the distribution of fluorescence of complement proteins or peptidoglycan along the length of *E. coli*. This study is an important addition to the understanding of how MAC pore formation leads to bacterial death.

Point 1: “The major unanswered question in this work is how MAC becomes localized to the division septum. Use of a fluorescently labeled anti-C5(b) antibody or fluorescently labeled C5 (if fluorophore is intact after C5 convertase cleavage) in C5-depleted serum would address whether this happens early in MAC assembly. It is interesting that although C8 mostly colocalizes with C9, it is less targeted to the pole/division septum than C9 (Figure 5). Could C9, with its membrane insertion domain, enable the targeting of MAC? This point could be mentioned in the Discussion (does not require an experiment).”

Answer 1: To address this question, we now added an extra figure showing the localization of C5 (Fig. EV5). C5 had a similar localization pattern as C9, suggesting that preferential MAC deposition is determined early in MAC assembly. However, since C5 does not have a membrane insertion domain, targeting of C5 could be caused by later MAC components, as also suggested by the referee for C8. C8 indeed seems slightly less targeted to the new pole compared to C9. Although C8 localizes at the new pole independently of C9, anchoring and polymerization of C9 may be more efficient at the new pole as well, which could lead to an even higher number of fully polymerized pores at these locations.

Point 2: “Does MAC deposition require full cell division and separation, or are the cell division-associated changes in PG or OM composition sufficient to drive MAC localization? Evaluating MAC in *E. coli* treated with PG inhibitors (like mecillinam, to cause filamentation) compared with inhibitors of bacterial growth patterning (like A22) would address this key question.”

Answer 2: We thank the reviewer for this valuable suggestion, which could provide interesting insights into the mechanisms underlying preferential MAC deposition at the new bacterial pole. In line with similar experiments that were suggested by referee #3, we now included new experiments with several different antibiotics to study the effect of active division, protein insertion and peptidoglycan synthesis on polar MAC deposition (Fig. EV4). These experiments suggest that active division and separation is not required for polar MAC deposition, but that the cell division-associated changes in the bacterial cell envelope are likely sufficient to drive MAC localization.

Minor comment 1: “Please describe in the Results section how the algorithm developed here accounts for out-of-focus fluorescence on the bacterium.”

Answer:

Out of focus bacteria are classified using our YOLO network and discarded for further processing during the analysis step. Generally, we only define a small subset (typically “Dividing” and “Rod” like bacteria) also detected by the classification, which are processed. Therefore, no out-of-focus fluorescence is used in our pipeline.

Minor comment 2: “In all figure legends, in addition to the number of images per experiment, please add the total number of bacteria that were analyzed. These numbers are a strength of the study.”

Answer: If analysis was performed, we now added the number of bacteria analyzed to the figure legends.

Minor comment 3: “In Figure 2, please explain why C1-inh was added for in vitro classical complement deposition.”

Answer: C1-inhibitor was added to prevent non-specific fluid-phase activation of C1, as described in the original paper about purified classical pathway system on bacteria (Muts *et al.*, Sci Rep, 2023). This is now also clarified in the results section.

Minor comment 4: “For Figure 3, please add a supplemental figure showing the change in location of HADA fluorescence in growing *E. coli* with pulse-chase.”

Answer: Pulse-chase labeling experiments with HADA were technically complicated to perform due to its weak signal and fast bleaching over time. However, we now added a figure following localization of HADA after different incubation times (Appendix Figure S7)

Minor comment 5: “The content of Movies EV1-EV4 is not defined; please add. If Figure EV6 is an image from one of the movies, indicate that in the figure legend.”

Answer: We added movie legends and clarified to which movie Figure EV6 belongs in the figure legend.

Minor comment 6: “Figure 4C: please make symbols bigger, or remake the graph to better emphasize the time-dependent changes for Sytox positivity and C9 position.”

Answer: We changed the symbols in this graph to make them easier to distinguish.

Referee #3:

In their manuscript, van't Wout et al. study the bactericidal action of the MAC complex using a reconstituted system consisting of purified and fluorescently labeled MAC components. This setup allows the authors to track the fate of live bacterial cells exposed to MAC complex formation in a reproducible and highly controllable manner. Using this powerful methodology, the authors find that MAC complexes localize to the septum or new pole of newborn *E. coli* cells at short exposure times, accompanied by rapid disruption of the inner bacterial membrane. To quantify this effect, they developed a sophisticated, deep learning assisted image analysis pipeline that allows for classification into dividing and non-dividing ("rod") cells and quantify the distribution of the fluorescence signal along the bacterial length axis. Using this pipeline, the authors study the dynamics of the MAC complex formation following different pathways, concluding that the localized formation at early time points might be targeting weak spots (areas of cell wall synthesis) of the cell. While the presented work is solid and observations are well described, it lacks mechanistical insight. There are several points that should be addressed to strengthen and verify the hypotheses presented in the manuscript. I hope that the comments are helpful to the authors.

Point 1: Underlying cell wall synthesis model

"The cell wall synthesis model employed by the authors and is based on polar PG synthesis. While recent studies have shown that polar PG is predominant during septation, PG insertion mainly occurs at the cell cylinder during cell elongation. This mode of cell wall synthesis is neither discussed by the authors nor taken into account during image analysis (neglection of "intermediate" cells, see next point). Cell elongation is more prominent during fast growth, while septation is predominant under slow growth conditions. The authors should provide information on the doubling time (either mass doubling time in OD measurements or single-cell doubling times from time-lapse images) to determine the predominant PG synthesis pathway."

Answer 1: We appreciate the comment of this reviewer. Based on the 0 min. timepoint video in figure 4, the average doubling time in our assay would be ~36 minutes. We think that analysis of the intermediate category, as proposed by this reviewer, was a very valuable suggestion to get more information about the underlying cell wall synthesis model. We further discuss this suggestion in the comment below.

Point 2: Classification of growth stages

"The authors classify individual bacteria into the categories "rod", "dividing", "intermediate", "defocused" and "microcolonies". The category "rod" should be changed to "non-dividing", as all cells are actually rod-shaped. Moreover, it is not clear to me, which criteria were applied to discriminate "rod" from "intermediate" cells. The authors state that only bacteria formed after cell division were classified as "rod-shaped", which limits the investigation of MAC formation to dividing and newborn cells. What happens with cells that "repaired" the weak spots after cell division, i.e. cells that were in the elongation phase when MAC components were added? Do these cells show less fluorescence? Quantifying fluorescence intensities for "intermediate" cells might provide important information with respect to the "weak spot" hypothesis. If only small cells were classified as "rod", the polar MAC distribution is not surprising, as it is a result of the completion of cell division events after MAC addition. In any case, the criteria for classification need to be specified and added to the manuscript."

Answer 2: We acknowledge that the naming of the categories is not fully accurate, as 'dividing' bacteria are still rod-shaped. Similarly, we think that 'dividing' is not fully correct either, as rod-shaped bacteria are also in the process of division. However, the naming and criteria that were used to distinguish the different bacterial growth stages are based on another paper (Spahn *et al.*,

Commun. Biol., 2022). ‘Rod-shaped’ bacteria were defined as the smallest bacteria that were recently separated, while ‘dividing’ bacteria needed to have a clear septum at mid-cell. These categories were validated by their length, as rod-shaped bacteria had to be smaller than dividing bacteria. Bacteria were classified as intermediate if they were already elongated but did not have a visible division septum yet. We especially thank the reviewer for their valid question about neglectation of the ‘intermediate’ category in our analysis. Analysis of the ‘intermediate’ category in figure 3 shows that these bacteria indeed show lower fluorescence intensities of C9, suggesting that MAC deposition is less efficient on bacteria that do not have these ‘weak spots’. We are now showing this analysis including an example image in Figure EV3.

Point 3: Quantification of fluorescence images

“The overall pipeline presented for single-cell analysis is innovative well designed. However, more information can be extracted from the data presented in the manuscript. The authors refer to the amount of MAC complexes in multiple sections of the manuscript without providing quantitative measures. As instance masks are available from the U-Net segmentation, it would be rather simple to quantify the integrated intensities per cell for the labeled subunits, providing a measure for the amount of pores. Such an analysis should be performed throughout the manuscript, as it allows the quantification of #pores vs time, #pores vs. cell size, #pores per growth stage, etc. I think that the recorded data is very interesting and that additional analyses could provide evidence for some of the postulated hypotheses. For example, on page 9 the authors state that the number of cell division correlated to the amount of MAC pores. However, there was no quantitative evidence provided other than a binary assessment (signal/no signal; growth/no growth), although the authors refer to Fig. EV6 for such an assessment. In the discussion (top lines on page 14), it is stated that bacteria with limited MAC deposition were able to recover. Here, proper quantification could provide a relative threshold of MAC pores that has to be reached for efficient killing.”

Answer 3: The image analysis pipeline presented in this study indeed enables us to provide a measure for the relative amount of pores. We now used this approach to make a correlation between the number of pores per bacterium and bacterial cell survival in Figure 4 (Appendix Figure S9). Since we can only directly compare bacteria within the same movie, we decided to use this analysis on the movie that was made after 6 minutes incubation with MAC components, as this sample contained the largest differences in bacterial survival.

Point 4: Novelty of the mechanism

“In the discussion, the authors state that polar MAC deposition was already observed in gram-positive bacteria in 2013. This raises the question about novelty of the observed phenotype. Can the authors highlight the advantages of their approach and discuss their results in the context of the mentioned study?”

Answer 4: The study that is mentioned by the referee describes the localization of C9 molecules on Gram-positive bacteria. These bacteria are intrinsically resistant to MAC-mediated killing and, most importantly, do not have an outer membrane in which MAC pores can insert. The C9 molecules that are detected are therefore most likely part of a non-polymerized and non-functional MAC pore. The current study therefore focuses on MAC-sensitive Gram-negative bacteria and describes the localization of biologically most relevant, lethal MAC pores, which we correlate with inner membrane damage. Next to that, in the study on Gram-positive bacteria, vastly differently localization patterns were found on different bacterial species. While C9 also localized near the division septum on *S. pyogenes*, they localized near the old poles of *B. subtilis* and *S. aureus* and more randomly on *L. lactis*. Due to the different cell envelope structure of Gram-positive bacteria, we believe that we are here describing an entirely different mechanism for C9 localization.

Point 5: Mechanism of polar/septal MAC pore formation

“The current work provides a solid description of MAC formation dynamics during early time points and its correlation with inner membrane damage. However, mechanistic insights into this process are hardly provided. For example, the authors state in the abstract that the “critical event causing MAC-mediated inner membrane damage remains elusive”. This question is not addressed by the current work. Another crucial question is why MAC pores form at the new cell pole. Can the authors speculate on this? Could lipid composition play a role? The authors suggest that the new pole represents a “weak spot”, but this hypothesis is only supported by similar patterns compared to nascent OMPs or septal PG synthesis. Regarding the observed localized MAC deposition, the authors pose several questions that could be tested directly. One question is whether active division is required for MAC-dependent killing. This question might be investigated using antibiotics that inhibit cell division (e.g. cephalexin). Another question is whether the increased insertion of OMPs at the division site enhances MAC deposition. This could be tested by inhibiting translation (e.g. using chloramphenicol) before and while adding MAC components. Another factor that could affect localization is cell curvature. Some proteins are known to be selective for positive or negative curvature (e.g. MreB, Min proteins) as curvature affects lipid bilayer packing. Could this also be a reason for the polar MAC deposition? The effect of curvature could be tested using antibiotics that affect cell elongation, such as A22 (MreB inhibitor) or Mecillinam (PBP2 inhibitor). It would be interesting to see if a morphological change affects MAC deposition.”

Answer 5: We thank the referee for these valuable suggestions, which can give interesting insights into the mechanism underlying polar MAC formation. We addressed this question through additional experiments, looking at MAC localization on bacteria that were treated with chloramphenicol (to inhibit protein synthesis), cephalexin (to inhibit peptidoglycan synthesis at mid-cell and cause filamentation), or mecillinam (to inhibit peptidoglycan synthesis at the lateral cell wall and make cells spherical). Similar localization patterns were observed on bacteria that were treated with peptidoglycan synthesis inhibitors, suggesting that active peptidoglycan synthesis is not required for preferential MAC localization. In addition, morphological changes do not seem to have a major effect on MAC localization either. Interestingly, however, less MAC deposition and Sytox influx were observed on bacteria treated with chloramphenicol, suggesting that active protein synthesis does influence the efficiency of MAC deposition.

Point 6: Are non-polar MAC pores required for killing?

“The CFU tests performed in this study showed that short incubation times that coincide with polar MAC deposition result in partial cell survival, while longer exposure with full deposition leads to complete killing. Isn't this an indication that non-polar deposition is at least partially contributing to killing?”

Answer 6: We agree that non-polar MAC pores are likely to at least partially contribute to bacterial killing. However, since we also already observe killing of most bacteria with polar MAC deposition, we believe that polar MAC pores can be sufficient to induce bacterial killing. The number of MAC pores is likely an important factor in determining bacterial survival, but the influence of their localization compared to the number of MAC pores would be more complicated to study.

Point 7: Killing in complement-activated cells

“The authors found that MAC-mediated killing takes place faster in complement-activated cells (Figure 2). While C9 is deposited earlier compared to non-IgM treated cells, inner membrane disruption dynamics are identical. This discrepancy is also seen in the representative cells (compare Fig. 1D vs. Fig. 2C). Do the authors have any explanation for this?”

Answer 7: This is indeed an interesting observation made by the reviewer. In the purified classical pathway system, C9 deposition starts faster than in the serum pre-opsonization assay, but there is a short delay of a few minutes before inner membrane damage occurs. Although we currently do not have an explanation for this, we hypothesize that there may be slightly more (pre-?) MAC pores that are not well inserted into the membrane in this purified assay, which do not trigger inner membrane damage.

Point 8: Statistics

“While CFU tests and flow cytometry experiments were conducted in biological replicates, while some imaging experiments were based on technical replicates (several images of one individual experiment). Can the authors comment on reproducibility of these experiments? Why didn't the authors perform these experiments in replicates?”

Answer 8: To validate the biological reproducibility of our findings all the experiments were performed three times. To also show this reproducibility within the manuscript, we now included overview images of these three independent biological replicates for the main figure of the manuscript in a supplementary figure (Appendix Figure S2). Besides this, we performed high-throughput analysis on a large number of images, and therefore bacteria from one data set per figure. This was done to validate that our observations made by eye were supported with a non-biased analysis method. The goal of this analysis was to robustly show that within one sample, the localization pattern was clearly present. This was crucial given that the samples are quite heterogeneous since they contain bacteria in different growth stages and MAC deposition can also slightly differ per bacterium within each sample.

Point 9: Selection of representative bacteria

“The manuscript only provides a few exemplary cells. To get a better impression of cellular heterogeneity, galleries of several cells for each condition/growth phase could be provided in the supplementary information. At best, these cells originate from different replicates to support the statement that the selected cells in the main text are "representative for the average cell".”

Answer 9: For each figure, we now included overview images in the appendix.

Point 10: Validation/quality control of the trained deep learning models

“The authors trained several deep learning models that were used for the image analysis. Typically, such models are tested in quality control experiments, in which their performance is quantified using suitable metrics. In case of segmentation and classification, this can be precision and recall or advanced metrics as dice index, F1-score or the panoptic quality. The authors should quantify model performance on unseen ground truth data and provide add the results to the manuscript.”

Answer 10:

We added metrics of the training process to the material and methods sections as well as training curves (Appendix Figure S10).

Point 11: Data availability

“The deep learning networks, trained models and the image analysis pipelines are very interesting. It would be great if the authors provide access to these resources by depositing them on suitable repositories (GitHub, Zenodo, etc.)”

Answer 11:

A github repository will be opened to the public upon publication including the models used, training data and raw data to replicate the analysis.

Point 12: Comprehensive model

“The experiments in this study give rise to MAC deposition dynamics and study the role of different pathways of the complement system. The findings/observations of this work could be well summarized in a comprehensive model. Such a model would assist the reader during the discussion and summarize the main points of this manuscript.”

Answer 12: We now added a figure showing our model for MAC deposition to the discussion (Fig. 6).

Other points:

- Page and line numbers should be added to simplify the review process
Page and line numbers are now added
- Introduction: What does "single membrane particles" mean?
We meant liposomes but wanted to emphasize the fact that they do not have two membrane like Gram-negative bacteria. We now changed this in the text.
- The time points (0, 2, 4 ... X min) refer to time when cells were sampled and proceeded for washing. How long were the centrifugation steps during washing? Don't the add to the treatment duration? Experimental conditions for washing should be added to the manuscript.
Centrifugation steps were performed by spinning down bacteria for 2 minutes at 10,000 xg.
- How was the number of cell divisions in Fig. 4 determined?
This was determined by counting the number of bacteria that were derived from a single bacterium after 2.5 hours. One was subtracted to get the number of divisions.
- The authors should provide the number of cells underlying the different experiments/conditions/time points in the figure caption.
The number of cells that was analyzed is now added to each figure legend.
- Time series shown in the supplementary videos could be stabilized (e.g. using the StackReg plugin in Fiji)
Time series are now corrected for sample drift using StackReg
- Scale bars and time stamps should be added to the supplementary videos.
Scale bars and time stamps have been added to the supplementary videos
- Was Sytox Blue also added for the control experiment in Movie EV?
Sytox is present in the agar pad. This information is now also added to the movie legend.
- The original U-Net paper by Ronnefeld et al. should be cited.
This paper is now added to the references.
- The data analysis is advanced and well-constructed. Thus it should be acknowledged more throughout the manuscript. Thank you for highlighting this, we have now included some words in the results section that acknowledge that the data analysis pipeline is advanced and innovative.

Answer: All these points are now addressed in the manuscript.

Dear Dr. Heesterbeek,

Thank you for the submission of your revised manuscript to our editorial offices. I have now received the reports from the two referees that I asked to re-evaluate the study (the points of referee #1 I considered as adequately addressed), you will find below. As you will see, referees #2 and #3 now support publication of your study in EMBO reports. Both have comments and suggestions to improve the manuscript, I ask you to address in a final revised manuscript. Please also provide a final p-b-p-response to the remaining referee points and the editorial requests below.

Editorial requests:

- Please indicate the corresponding author on the title page and add an e-mail contact.
- Please add up to five keywords to the manuscript, below the abstract.
- Please provide the abstract written in present tense throughout.
- We now use CRediT to specify the contributions of each author in the journal submission system. CRediT replaces the author contribution section. Please use the free text box to provide more detailed descriptions and do NOT provide your final manuscript text file with an author contributions section. See also our guide to authors:
<https://www.embopress.org/page/journal/14693178/authorguide#authorshipguidelines>
- Please remove the referee access from the data availability section (DAS) and add direct, publicly accessible links. Please make sure that datasets and codes are public latest upon online publication of the manuscript.
- Please use our reference format:
<http://www.embopress.org/page/journal/14693178/authorguide#referencesformat>
- The nomenclature of the four movie files is wrong and needs to be corrected to Movie EVx in all places (source file names, titles in the system, legends, callouts in the manuscript text). Please remove the movie legends the manuscript text file. Each legend needs to be provided as a separate text file ZIPped together with its movie so that we have one ZIP folder per movie that is uploaded.
- Please check again that the number "n" for how many independent experiments were performed, their nature (biological versus technical replicates), the bars and error bars (e.g. SEM, SD) and the test used to calculate p-values is indicated in the respective figure legends (main, EV and Appendix figures). Please also check that all the p-values are explained in the legend, and that these fit to those shown in the figure. Please provide statistical testing where applicable. Please avoid the phrase 'independent experiment' but clearly state if these were biological or technical replicates. Please also indicate (e.g. with n.s.) if testing was performed, but the differences are not significant. In case n=2, please show the data as separate datapoints without error bars and statistics. See also:
<http://www.embopress.org/page/journal/14693178/authorguide#statisticalanalysis>
- If n<5, please show single datapoints for diagrams. Could statistics be provided for the diagrams shown in panels 4A, 4B and EV4A? The phrase 'independent experiment' is still used several times. Please replace this and clearly state if these were biological or technical replicates. Moreover:
 - Please note that the exact p values are not provided in the legend of figure 4C.
 - Please note that the box plots need to be defined in terms of minima, maxima, centre, bounds of box and whiskers, and percentile in the legends of figures 1E, F; 2D, 3D, 5B, D, F; EV1 D, EV2 B, EV3 C, EV5B.
 - Please note that the error bars are not defined in the legend of figure EV4 A.
- Please add to each legend (main and Appendix figures, where applicable) a 'Data Information' section explaining the statistics used or providing information regarding replicates and scales. See:
<https://www.embopress.org/page/journal/14693178/authorguide#figureformat>
- Please remove the instructions from the Reagents & Tools Table.
- Please add the title of the paper to the Appendix title page ('Appendix for ...').
- Thanks for providing the source data. Please upload this as one folder per main figure, grouping together all the files for this figure (and ZIPped together), one folder for the EV figures, grouping together all the files for each EV figure in separate folders (and ZIPped together), and one folder for the Appendix figures.

In addition, I would need from you uploaded separately (please remove this from the manuscript text file):

Best,

Referee #2:

The authors have done a commendable job responding to all reviewers' points. The results with antibiotics are a strong addition to the manuscript.

My only point is regarding lines 207-209, and response to reviewer #3. Appendix Fig. S9 B-C suggest there is a threshold of C9 or Sytox positivity, after which cell division is compromised. The data could be subjected to re-analysis, placing a threshold for the regression line (around $x = 9$ in B and $x = 27$ in C). This possibility for a threshold of intensity that correlates with restricted bacterial replication could be mentioned in Discussion in the paragraph ending with lines 307-312.

Referee #3:

In the revised manuscript, van't Wout and colleagues extensively significantly expanded their investigation of localized MAC deposition onto the surface of *E. coli* cells. They conducted additional experiments that cover the questions raised by the reviewers. The combination of MAC and antibiotic treatments adds significant value and allowed the authors to speculate more about the mechanisms underlying the complement-mediated killing. The new results and analyses strengthen the weak spot hypothesis provided by the authors. Next to the additional experimental work, the method section was expanded significantly, improving the reproducibility and providing useful information to the community. In summary, the work by van't Wout and coworkers both provides novel and interesting insights into MAC-mediated killing and an experimental and analytical framework that is valuable for further experiments. The image analysis is state of the art and will be useful for the community.

Nevertheless, there are a few points that might want to be considered before publication of the article.

1. I acknowledge both the annotation guidelines and the quantification of model performance, as it is now provided in the manuscript. However, the performance is not discussed in the main text. Could the authors briefly elaborate, how a classification accuracy of 0.7 can affect the results? Was there a selection criterium for the automated analysis to reduce the effect of false classification?
2. In line 154, the authors still mention that PG synthesis mainly occurs at the cell pole. As mentioned before, this is only true during septation, as cell elongation takes place along the cylindrical part. This is not as prominently visible in the HADA pulse-chase experiment, as it represents a dilution of the PG (new precursors are integrated into the existing cell wall), while septal PG is synthesized *de novo*. I suggest that the authors add "during septation/cell division", as they did later in the manuscript.
3. The contrast in many figures is quite low (Figures S2, S5, S8, S10, EV figures 4 and 5). The contrast should be adjusted at least for EV figures 4 and 5, as cells are hardly visible in the phase contrast channel.
4. In their R2R, the authors mention that the nomenclature for cell classification were inspired by the work of Spahn et al.. As this work seems to be relevant, it might be added as a reference to the manuscript.
5. The drug treatment experiments strengthened the hypothesis that weak spots in septal PG recruit the MAC. However, it is not clear how these weak spots affect the outer membrane and lead to the recruitment. The authors mention that lipid composition and packing could be a reason. Is there any hints from literature that could support this hypothesis. Also, the results of cephalixin treatment seems to be highly interesting. Although there is no septation, MAC complexes appear to integrate at future division sites, and the signal appears to spread more rapidly than in untreated cells. This observation is not picked up in

the manuscript so far. In contrary, the authors mention that the treatment does not change the localization pattern (lines 281-283), although the recruitment differs from untreated cells (deposition at the cylindrical part and not to the septum). This effect might even be quantified with the current analysis pipelines.

6. In lines 271 - 278, the authors speculate on the mechanism underlying the localized deposition of membrane attack complexes. The described phenotype of LPS and Cardiolipin localization speaks against these molecules as molecular determinants. This, however, is not mentioned directly. The authors should conclude this discussion with a more direct statement.

7. It is great that the authors provided a model as Figure 6. However, this model does not include many of the observations. Examples are membrane damage, which occurs concomitantly with MAC deposition. The spread of MAC signal over the entire cell surface over extended incubation times is not immediately graspable. The arrow size could be increased, the arrows labeled or another time point with overall labeling of the membrane could be included. The time-resolved measurements in this work would further allow to add temporal information to the model. Finally, the weak-spot hypothesis could be added to the model, as it is the main interpretation of the extensive data provided in this manuscript.

Minor points:

Line 61: „It is, however, unknown where...”

Line 64: "Here, ..."

Line 66: Due to the short time delay, the authors might consider to use "correlates" instead of "coincides".

Revised manuscript of “**Bactericidal Membrane Attack Complex formation initiates at the new pole of *E. coli***” – reference EMBOR-2025-61837-T

Utrecht, September 19th, 2025

Dear Achim Breiling, dear editors of EMBO Reports,

Hereby we send you our final revised manuscript entitled “**Bactericidal Membrane Attack Complex formation initiates at the new pole of *E. coli***”. The revised manuscript addresses all editorial request and the final comments of referee #2 and #3. In the point-to-point reply below, we answer the points that were addressed by the reviewers and indicate how we have included these comments in the manuscript.

We trust that our revised version of the manuscript is now suitable for publication in EMBO Reports.

Sincerely, on behalf of the authors,
Dr. Dani Heesterbeek
University Medical Center Utrecht
Medical Microbiology, G04.614
Heidelberglaan 100, 3584 CX
Utrecht, The Netherlands
d.a.c.heesterbeek-2@umcutrecht.nl

POINT-TO-POINT REPLY II

Referee #2:

The authors have done a commendable job responding to all reviewers' points. The results with antibiotics are a strong addition to the manuscript.

Point 1: My only point is regarding lines 207-209, and response to reviewer #3. Appendix Fig. S9 B-C suggest there is a threshold of C9 or Sytox positivity, after which cell division is compromised. The data could be subjected to re-analysis, placing a threshold for the regression line (around $x = 9$ in B and $x = 27$ in C). This possibility for a threshold of intensity that correlates with restricted bacterial replication could be mentioned in Discussion in the paragraph ending with lines 307-312.

Answer 1: We now performed re-analysis of these graphs, placing a threshold for the regression line at a maximum C9 intensity of 9 and a maximum Sytox intensity of 28 (see D,E in figure below). We agree that there seems to be a threshold of intensity that correlates with restricted growth, but we think there is not enough data to make this correlation. It is also not an absolute threshold, as some bacteria can still grow after this threshold (although less efficiently). Therefore, we think our data are not strong enough to state that there is a threshold correlating with compromised cell division, and we decided to not change the figure in our manuscript.

Data information: In (A), slope = 0.83, R = 0.82, R² = 0.67, p < 0.0001. In (B), slope = -0.47, R = -0.70, R² = 0.48, p = 0.0019. In (C), slope = -0.43, R = -0.65, R² = 0.42, p = 0.0047. In (D), slope = -0.85, R = -0.62, R² = 0.38, p = 0.0757. In (E), slope = -1.34, R = -0.69, R² = 0.48, p = 0.0399.

Referee #3:

In the revised manuscript, van 't Wout and colleagues extensively significantly expanded their investigation of localized MAC deposition onto the surface of *E. coli* cells. They conducted additional experiments that cover the questions raised by the reviewers. The combination of MAC and antibiotic treatments adds significant value and allowed the authors to speculate more about the mechanisms underlying the complement-mediated killing. The new results and analyses strengthen the weak spot hypothesis provided by the authors. Next to the additional experimental work, the method section was expanded significantly, improving the reproducibility and providing useful information to the community. In summary, the work by van't Wout and coworkers both provides novel and interesting insights into MAC-mediated killing and an experimental and analytical framework that is valuable for further experiments. The image analysis is state of the art and will be useful for the community.

Nevertheless, there are a few points that might want to be considered before publication of the article.

Point 1: I acknowledge both the annotation guidelines and the quantification of model performance, as it is now provided in the manuscript. However, the performance is not discussed in the main text. Could the authors briefly elaborate, how a classification accuracy of 0.7 can affect the results? Was there a selection criterium for the automated analysis to reduce the effect of false classification?

Answer 1: We are convinced that our model is reliable, as the mean average precision is considered high for a YOLO model. For instance, for a medium YOLO model with 81 classes, mAP50-95 is 51.5 (<https://docs.ultralytics.com/models/yolo11/#performance-metrics>). Since we did observe some misclassified bacteria (rod shaped vs dividing) during the average shape determination, we added an additional length criterium in the python notebooks (automatic Otsu threshold of the bacterial length) to mitigate misclassification from the YOLO classifier. We now also discuss this in the results section of the main text.

Point 2: In line 154, the authors still mention that PG synthesis mainly occurs at the cell pole. As mentioned before, this is only true during septation, as cell elongation takes place along the cylindrical part. This is not as prominently visible in the HADA pulse-chase experiment, as it represents a dilution of the PG (new precursors are integrated into the existing cell wall), while septal PG is synthesized de novo. I suggest that the authors add "during septation/cell division", as they did later in the manuscript.

Answer 2: We now added 'during cell division' to this sentence to highlight that peptidoglycan synthesis primarily occurs at the septum only during cell division, as it occurs along the cylindrical part during elongation.

Point 3: The contrast in many figures is quite low (Figures S2, S5, S8, S10, EV figures 4 and 5). The contrast should be adjusted at least for EV figures 4 and 5, as cells are hardly visible in the phase contrast channel.

Answer 3: The contrast of the phase-contrast channels in figure EV4 and figure EV5 is now adjusted to increase the visibility of the bacteria.

Point 4: In their R2R, the authors mention that the nomenclature for cell classification were inspired by the work of Spahn et al.. As this work seems to be relevant, it might be added as a reference to the manuscript.

Answer 4: We agree and added this reference to the manuscript.

Point 5: The drug treatment experiments strengthened the hypothesis that weak spots in septal PG recruit the MAC. However, it is not clear how these weak spots affect the outer membrane and lead to the recruitment. The authors mention that lipid composition and packing could be a reason. Is there any hints from literature that could support this hypothesis. Also, the results of cephalexin treatment seems to be highly interesting. Although there is no septation, MAC complexes appear to integrate at future division sites, and the signal appears to spread more rapidly than in untreated cells. This observation is not picked up in the manuscript so far. In contrary, the authors mention that the treatment does not change the localization pattern (lines 281-283), although the recruitment differs from untreated cells (deposition at the cylindrical part and not to the septum). This effect might even be quantified with the current analysis pipelines.

Answer 5: We also think that the results of cephalexin treatment are very interesting. We now discuss this more elaborately in the results and discussion section. It would be great if we could quantify the effect as well, but since these bacteria have a completely different shape compared to untreated bacteria, that would require us to re-train our deep learning model. This will be something we could do for a future study.

Point 6: In lines 271 - 278, the authors speculate on the mechanism underlying the localized deposition of membrane attack complexes. The described phenotype of LPS and Cardiolipin localization speaks against these molecules as molecular determinants. This, however, is not mentioned directly. The authors should conclude this discussion with a more direct statement.

Answer 6: We now state more clearly that LPS and cardiolipin are unlikely to determine localized MAC deposition.

Point 7: It is great that the authors provided a model as Figure 6. However, this model does not include many of the observations. Examples are membrane damage, which occurs concomitantly with MAC deposition. The spread of MAC signal over the entire cell surface over extended incubation times is not immediately graspable. The arrow size could be increased, the arrows labeled or another time point with overall labeling of the membrane could be included. The time-resolved measurements in this work would further allow to add temporal information to the model. Finally, the weak-spot hypothesis could be added to the model, as it is the main interpretation of the extensive data provided in this manuscript.

Answer 7: We adjusted our comprehensive model to include more of the information described in the manuscript.

Minor points:

Line 61: „It is, however, unknown where...”

Line 64: "Here, ..."

Line 66: Due to the short time delay, the authors might consider to use "correlates" instead of "coincides".

We changed these sentences in the manuscript.

Dr. Dani Heesterbeek
University Medical Center Utrecht
Medical Microbiology
Heidelberglaan 100, G04.614
Utrecht 3584CX
Netherlands

Dear Dr. Heesterbeek,

Thank you for the submission of your final revised manuscript to EMBO reports. I now went through this and consider the remaining referee points as adequately addressed.

I am thus very pleased to accept your manuscript for publication in the next available issue of EMBO reports. Thank you for your contribution to our journal.

Yours sincerely,
